# ENHANCING MULTIVARIATE TIME SERIES FORECAST-ING WITH GLOBAL TEMPORAL RETRIEVAL

**Fanpu Cao[1], Lu Dai[1,2], Jindong Han[3], Hui Xiong[1,2]***

[1]The Hong Kong University of Science and Technology (Guangzhou), Guangzhou, China
[2]The Hong Kong University of Science and Technology, Hong Kong SAR, China
[3]Shandong University, Jinan, China
`fcao628@connect.hkust-gz.edu.cn; ldaiae@connect.ust.hk;`
`jindong.han@sdu.edu.cn; xionghui@ust.hk`

## ABSTRACT

Multivariate time series forecasting (MTSF) plays a vital role in numerous real-world applications, yet existing models remain constrained by their reliance on a limited historical context. This limitation prevents them from effectively capturing global periodic patterns that often span cycles significantly longer than the input horizon—despite such patterns carrying strong predictive signals. Naïve solutions, such as extending the historical window, lead to severe drawbacks, including over-fitting, prohibitive computational costs, and redundant information processing. To address these challenges, we introduce the Global Temporal Retriever (GTR), a lightweight and plug-and-play module designed to extend any forecasting model's temporal awareness beyond the immediate historical context. GTR maintains an adaptive global temporal embedding of the entire cycle and dynamically retrieves and aligns relevant global segments with the input sequence. By jointly modeling local and global dependencies through a 2D convolution and residual fusion, GTR effectively bridges short-term observations with long-term periodicity without alter-ing the host model architecture. Extensive experiments on six real-world datasets demonstrate that GTR consistently delivers state-of-the-art performance across both short-term and long-term forecasting scenarios, while incurring minimal parameter and computational overhead. These results highlight GTR as an efficient and general solution for enhancing global periodicity modeling in MTSF tasks. Code is available at this repository: https://github.com/macovaseas/GTR.

## 1 INTRODUCTION

Multivariate time series forecasting (MTSF) is a critical task with widespread applications in numer-ous domains, including energy grid management (Alvarez et al., 2010), climate modeling (Mudelsee, 2010), macroeconomic planning (Granger & Newbold, 2014) and traffic flow management (Ishak & Al-Deek, 2002). Accurate forecasting in these areas is essential for resource optimization, strategic planning, and risk mitigation. In recent years, deep learning-based methods have achieved state-of-the-art performance in MTSF tasks, with a variety of architectures including MLPs (Das et al., 2023), RNNs (Lin et al., 2025c), CNNs (Liu et al., 2022a), Transformers (Zhou et al., 2021) and Mambas (Zhang et al., 2023) demonstrating strong performance.

A fundamental challenge in time series forecasting lies in effectively modeling periodicity. Real-world time series data often exhibit complex periodic patterns at multiple scales, which can be broadly categorized into *local* and *global* cycles (Wang et al., 2024c). Local periodic patterns, such as daily fluctuations, are characterized by their high frequency and distinct, repetitive nature, making them easy for models to capture from a limited historical window. In contrast, global periodic patterns—like weekly, monthly, or seasonal trends—present a more significant challenge. These patterns gradually unfold over long time spans, occur less frequently within look-back window, and are often obscured by non-stationary phenomena such as extreme values, missing data, and noisy perturbations (Liu et al., 2022b). Despite these difficulties, effectively capturing global periodic

---

*Corresponding author.

information is crucial, as the predictive signal from a global cycle can be much stronger than that from local, adjacent patterns. As illustrated in Figure 1, the correlation between a time segment and its distant counterpart in the global cycle is often higher than its correlation with its immediate neighbors. This demonstrates that global dependencies can be more informative for forecasting than adjacent temporal data, making them essential to harness for achieving high-accuracy predictions.

To capture complex periodic dynamics, researchers have explored several strategies. One prominent line of work employs seasonal-trend decomposition, isolating periodic components from the series for specialized modeling (Wu et al., 2021; Zhou et al., 2022a; Zeng et al., 2023; Wang et al., 2024c; 2025a). Another strategy operates in the frequency domain, using tools such as the Fast Fourier Transform (FFT) (Brigham & Morrow, 1967) to capture cyclical signals more explicitly (Zhou et al., 2022b; Xu et al., 2024; Yi et al., 2024). A third stream focuses on reshaping data representations; for instance, Times-Net (Wu et al., 2022) transforms 1D sequences into 2D tensors to better capture both intra-period and inter-period variations. These approaches have led to notable progress, but they all share a fundamental limitation: they operate strictly within a fixed look-back window. As a result, when the true cycle length extends far beyond the observed history, global periodic patterns remain invisible to the model.

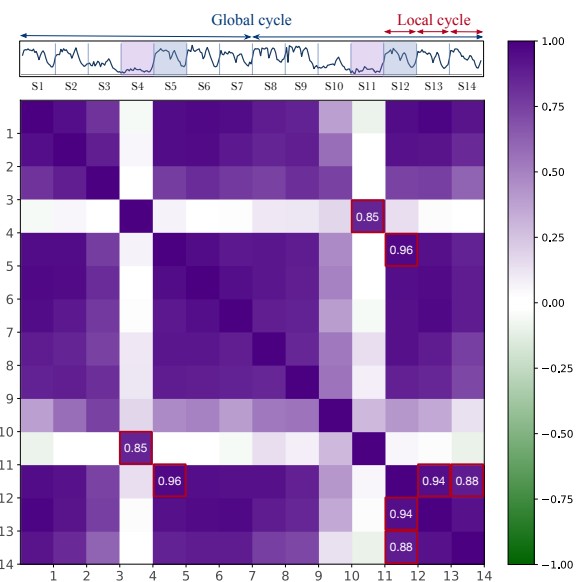

Figure 1: Pearson correlation matrix of time series segments from the Electricity dataset. We divided the series into several sub-series by the local-cycle length. The series demonstrates both the local-cycle pattern (e.g., $Corr(S_{12}, S_{13}) = 0.94, Corr(S_{12}, S_{14}) = 0.88$) and global-cycle pattern (e.g., $Corr(S_{12}, S_5) = 0.96$). Critically, the global-cycle pattern is stronger than local-cycle pattern (e.g., $Corr(S_{12}, S_5) > Corr(S_{12}, S_{13}), Corr(S_{12}, S_{14})$).

A straightforward remedy is to simply extend the look-back window, but this approach quickly proves impractical for several interconnected reasons (Tong & Yuan, 2025). First, as the input length grows, models risk overfitting to spurious noise, which undermines forecast stability rather than improving it. Second, longer input windows substantially inflate both computational and memory costs, often exceeding practical resource budgets. Finally, even with abundant history available, extracting truly relevant signals from overwhelming amounts of data remains inherently difficult, as models must distinguish meaningful long-term patterns from redundant or irrelevant information. Thus, capturing global periodicity requires a more principled solution than brute-force window extension.

To address this issue, we introduce the Global Temporal Retriever (GTR), a lightweight and plug-and-play module designed to extend a forecaster's temporal awareness beyond the immediate historical context. At its core, GTR maintains an adaptive global temporal embedding that encodes the long-range periodic structures across the entire cycle. For each input sequence, GTR first identifies its absolute position within the global cycle and retrieves the corresponding temporal segment from the learned global embedding. The retrieved segment is then aligned with the local input and fused through a lightweight 2D convolution, enabling joint modeling of both local-cycle dynamics and global-cycle dependencies. The resulting enriched representation is finally integrated back into the forecasting backbone via a residual connection, enhancing predictive power without altering its original architecture. Extensive experiments demonstrate that combining GTR technique with a simple MLP backbone achieves state-of-the-art performance on 6 real-world multivariate datasets by effectively capturing long-cycle dependencies. Importantly, GTR is highly efficient, introducing negligible parameter and runtime overhead, and can be seamlessly integrated into a wide range of forecasting backbones.

The primary contributions of this work are as follows:

- We pinpoint a core limitation in current MTSF methods: fixed look-back windows often obscure global periodic patterns whose cycles exceed the observed history.

- We present the Global Temporal Retriever (GTR), a lightweight, plug-and-play, model-agnostic module that uses absolute temporal indexing to retrieve from an adaptive global cycle embedding and fuses local and global cues via 2D convolution—requiring no changes to the host forecaster.

- Across six benchmarks and both long- and short-term settings, GTR consistently improves diverse forecasting backbones and achieves overall state-of-the-art accuracy with minimal parameter and runtime overhead.

## 2 RELATED WORK

Multivariate time series forecasting (MTSF) has widespread applications in various domains. In recent years, deep learning-based methods have achieved great success in MTSF tasks, including RNN-based methods (Lin et al., 2025c; Bergsma et al., 2023; Wang et al., 2024b), CNN-based methods (Liu et al., 2022a; Wu et al., 2022; Luo & Wang, 2024), Linear & MLP-based methods (Zeng et al., 2023; Das et al., 2023; Lin et al., 2024), Transformer-based methods (Nie et al., 2023; Liu et al., 2024b; Wang et al., 2024d) and Mamba-based methods (Zhang et al., 2023; Wang et al., 2025b; Ma et al., 2025). Among these methods, *explicit periodicity modeling* and *2D structural transformation* have rapidly evolved as dominant paradigms, with the former focusing on extracting multi-scale temporal patterns and the latter leveraging 2D representations to capture complex patterns beyond conventional sequential modeling.

**Modeling Periodicity of Time Series.** Modeling temporal periodicity has long been used to boost forecasting accuracy. Recent methods such as Autoformer (Wu et al., 2021), FEDformer (Zhou et al., 2022a), DLinear (Zeng et al., 2023), and the TimeMixer family (Wang et al., 2024c; 2025a) perform seasonal–trend decomposition to better capture periodic components in the original time series. In parallel, frequency-domain modeling has been increasingly integrated into deep networks: FiLM, FITS, and FilterNet use FFT-based representations to capture patterns that are hard to express in the time domain (Zhou et al., 2022b; Xu et al., 2024; Yi et al., 2024; Brigham & Morrow, 1967). CycleNet further learns recurrent cycles as explicit periodic structures (Lin et al., 2024). These methods effectively exploit multi-scale periodicity but are often bounded by the observed window or by assumptions of stationary cycles, limiting mining of information in very long cycles.

**2D-Variations of Time Series Modeling.** Transforming 1D time series data into a 2D representation allows models to capture more complex features that are difficult for traditional sequential methods to extract. TimesNet (Wu et al., 2022) captures the temporal 2D-variations in 2D space by CNN-based vision backbones for the first time. LightTS (Zhang et al., 2022) leverages two distinct sampling strategies to structure the 2D time-series data and employs MLPs for efficient feature extraction. MDCNet (Su et al., 2024) employs multi-scale 2D convolutional networks on variational mode-decomposed time series components to extract multi-frequency patterns. Times2D (Nematirad et al., 2025) converts 1D time series into 2D tensors via frequency-domain periodic decomposition and derivative heatmaps, leveraging 2D convolutions to capture short- and long-term dependencies. Clustering-enhanced modeling such as DUET introduces dual clustering on temporal and channel dimensions so as to capture both intra-series patterns and inter-channel dependencies (Qiu et al., 2025). Despite stronger representations, these approaches infer periodic structure indirectly from the re-layout of data and still lack a consistent mechanism to align with global cycles.

**Retrieval-/Memory-Augmented Forecasting.** Retrieval-augmented methods enlarge the effective context by fetching similar segments from the full history or external repositories and fusing them into the predictor (Liu et al., 2024a; Han et al., 2025; Ning et al., 2025; Yang et al., 2025b). Such designs improve generalization and few-shot forecasting by exposing the model to recurring patterns beyond the current window. However, they rely heavily on similarity search quality and do not by themselves provide a compact, time-aligned representation of very long cycles, motivating approaches that directly encode and align global periodicity while remaining backbone-agnostic.

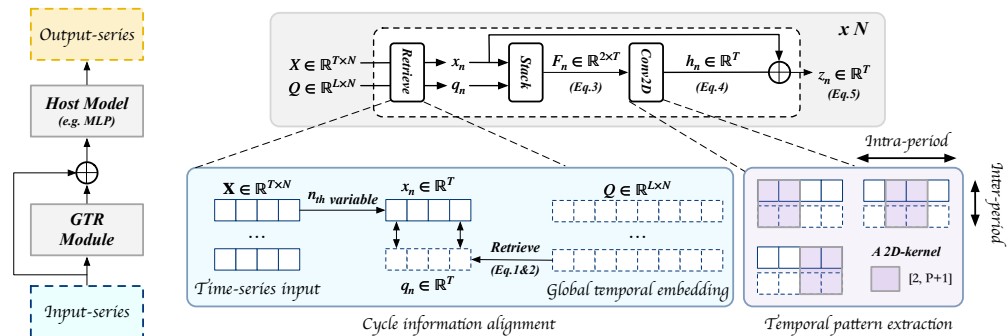

Figure 2: Overview of the Global Temporal Retriever (GTR): a plug-and-play module compatible with any MTSF forecaster. GTR operates in three stages: (1) retrieves corresponding segments from global temporal embedding; (2) aligns them with the input and uses 2D convolution to jointly model local and global periodicity; (3) fuses the result with the original input via residual connection.

# 3 METHOD

## 3.1 PROBLEM STATEMENT AND OVERVIEW

In multivariate time series forecasting, given historical observations $\boldsymbol{X} = \{x_1, \ldots, x_T\} \in \mathbb{R}^{T \times N}$, where $T$ represents the number of time steps and $N$ represents the number of variables, our target is predicting the future $S$ time steps $\boldsymbol{Y} = \{x_{T+1}, \ldots, x_{T+S}\} \in \mathbb{R}^{S \times N}$. Critically, we focus on scenarios where the global cycle length significantly exceeds the historical input length $T$ — a common yet challenging setting in real-world time series forecasting.

**Method Overview.** In this paper, we propose the *Global Temporal Retriever (GTR)* — a lightweight, plug-and-play module designed to extend a model's temporal receptive field beyond the immediate input window. As illustrated in Figure 2, the proposed method operates in two phases: (1) The GTR module enhances global cyclic patterns by dynamically retrieving periodic information from the global temporal embedding, then fusing them with the input series through a linear transformation and 2D convolution (c.f. Section 3.2). (2) The enhanced representation is subsequently processed by the backbone model (a multi-layer perceptron in this work, c.f. Section 3.3.) for final forecasting.

## 3.2 GLOBAL TEMPORAL RETRIEVER

To address the fundamental limitation of lookback window constraints in capturing global periodic patterns, we propose the Global Temporal Retriever (GTR), a lightweight and plug-and-play module that enables models to access temporal information beyond the immediate historical context.

The core innovation of GTR lies in maintaining an adaptive global temporal embedding that encodes the entire cycle pattern of the time series. Specifically, we introduce a global parameter matrix $\boldsymbol{Q} \in \mathbb{R}^{L \times N}$, where $L$ denotes the global cycle length and $N$ represents the number of variables. This parameter is initialized to zero and automatically learns to capture the global periodic patterns inherent in each variable in the time series during training.

For notational convenience, we consider the univariate case. Given the $n$-th variable sequence $\mathbf{x}_n \in \mathbb{R}^T$ with lookback length $T$, the GTR module operates through two principled stages to bridge local observations with global temporal structure:

**Cycle Information Alignment.** To establish continuity with the global temporal structure, we define a cycle index vector $\boldsymbol{i} \in \mathbb{N}_0^T$ that precisely locates each position within the global cycle. For a sequence starting at absolute time $t_0$, the index vector is computed as:

$$\boldsymbol{i} = \left[(t_0 \bmod L) + \tau\right] \bmod L \quad \text{for} \quad \tau = 0, 1, \ldots, T - 1. \tag{1}$$

The cycle index enables positional alignment of the input sequence within the global periodic range. Using the index vector, we retrieve corresponding temporal references from the global cycle representation:

$$\boldsymbol{q}_n = \text{Linear}(\boldsymbol{Q}[\boldsymbol{i}, n]) \in \mathbb{R}^T, \tag{2}$$

where $\boldsymbol{q}_n$ represents the retrieved global temporal information corresponding to the current sequence's position within the cycle. The linear mapping is utilized to enhance the representational capacity of the temporal reference. We then stack the original input sequence and the aligned global query to facilitate interaction between local patterns and global cycle information:

$$\boldsymbol{F}_n = \begin{bmatrix} \boldsymbol{x}_n \\ \boldsymbol{q}_n \end{bmatrix} \in \mathbb{R}^{2 \times T} \tag{3}$$

**Temporal Pattern Extraction.** We apply a 2D convolution operation to extract temporal patterns that span both local and global scales:

$$\mathbf{h}_n = \mathcal{C}(\boldsymbol{F}_n; \kappa = (2, 1 + 2\lfloor P/2 \rfloor)) \in \mathbb{R}^T \tag{4}$$

where $\mathcal{C}$ denotes the convolution operator, and $\kappa$ is designed with $P$ representing the dominant high-frequency period length (e.g., daily patterns in hourly data), capturing interactions across the two temporal scales while preserving the periodic structure. Finally, the extracted features are integrated with the original sequence through a residual connection with dropout:

$$\mathbf{z}_n = \mathbf{x}_n + \text{Dropout}(\mathbf{h}_n) \in \mathbb{R}^T \tag{5}$$

The key advantage of GTR is its ability to enable the model to retrieve and utilize periodic information from the entire cycle, regardless of the lookback window length. This is particularly beneficial for time series with long-term periodicities (*e.g.*, monthly or yearly patterns) that cannot be captured within typical short lookback windows.

## 3.3 BACKBONE AND PROJECTION STRUCTURE

Many state-of-the-art methods in multivariate time series forecasting (MTSF) are built upon Multi-Layer Perceptrons (MLPs) (Das et al., 2023; Lin et al., 2024), demonstrating strong empirical performance and robustness. Inspired by these successes, we adopt a lightweight MLP-based backbone to capture temporal dependencies in the sequence.

**Input Projection.** Let $\boldsymbol{Z} \in \mathbb{R}^{T \times N}$ be the representation derived from the GTR Technique, we first project the representation into a higher-dimensional latent space via a linear transformation:

$$\boldsymbol{Z} = \text{Linear}(\boldsymbol{Z}) \in \mathbb{R}^{D \times N}.$$

**Multi-Layer Perceptron.** The MLP backbone consists of two linear layers with GeLU activation functions (Hendrycks & Gimpel, 2023) and a residual connection. Formally:

$$\boldsymbol{G}_1 = \text{GeLU}(\text{Linear}(\boldsymbol{Z})) \in \mathbb{R}^{D \times N},$$
$$\boldsymbol{G}_2 = \text{GeLU}(\text{Linear}(\boldsymbol{G}_1)) \in \mathbb{R}^{D \times N},$$
$$\boldsymbol{Z}_{\text{out}} = \boldsymbol{G}_2 + \boldsymbol{Z} \in \mathbb{R}^{D \times N}.$$

**Output Projection.** The extracted features are then mapped to the forecast horizon via a linear output layer, applied after a dropout layer for regularization:

$$\hat{\boldsymbol{Y}} = \text{Linear}(\text{Dropout}(\boldsymbol{Z}_{\text{out}})) \in \mathbb{R}^{S \times N}.$$

## 3.4 METHOD DETAILS

**Instance Normalization.** Distribution shift between training and testing data is a prevalent challenge in time series forecasting (Liu et al., 2022b). Reversible Instance Normalization (RevIN) (Kim et al., 2021) has been demonstrated to effectively alleviate this issue by applying instance normalization and denormalization to the inputs and outputs of the model. The module normalizes the input batch by removing instance-specific mean and variance, and subsequently reverses this operation on the model outputs. We adopt RevIN to stabilize the non-stationarity of the input time series.

**Seamless Compatibility.** GTR preserves input dimensionality while integrating global periodic patterns, enabling plug-and-play integration with any forecasting backbone without architectural modifications. It processes raw input series to generate enhanced feature representations, directly fed into the backbone for end-to-end training.

**Complexity Analysis.** Reversible instance normalization operates at $O(NT)$ complexity. Series embedding requires $O(NTd)$ operations due to projection into the hidden dimension. The GTR module's linear mapping dominates with $O(NT^2)$ complexity, while the 2D convolution is negligible since cycle length $P$ is a small constant. The backbone and predictor exhibit $O(Nd^2)$ and $O(NSd)$ complexity, respectively. The total complexity is $O(NT^2 + Nd^2 + NTd + NSd)$, which is linear to $N$, and $S$. When $T$ is significantly smaller than $d$, the complexity is dominated by the $O(Nd^2)$ term.

## 4 EXPERIMENT

### 4.1 EXPERIMENT SETUP

**Dataset.** We evaluate the proposed method on six widely used real-world datasets, including the ETT series (Zhou et al., 2021), PEMS series (Liu et al., 2024b), Electricity, Traffic, Solar-Energy and Weather datasets (Wu et al., 2021). A detailed description of this part is provided in Appendix A.1.

**Evaluation.** We use Mean Squared Error (MSE) and Mean Absolute Error (MAE) as the core metrics for the evaluation. To fairly compare the forecasting performance, we follow the same evaluation protocol, where the length of the historical horizon is set as $T = 96$ for all models. We set the prediction lengths $S$ to $\{12, 24, 48, 96\}$ for PEMS dataset and $\{96, 192, 336, 720\}$ for others.

**Baselines.** To evaluate the performance of the proposed method, we compare it against several recently well-acknowledged forecasting models, including RAFT (Han et al., 2025), S-Mamba (Wang et al., 2025b), TQNet (Lin et al., 2025a), TimeXer (Wang et al., 2024d), CycleNet (Lin et al., 2024), SOFTS (Han et al., 2024), TimeMixer (Wang et al., 2024c), iTransformer (Liu et al., 2024b), PatchTST (Nie et al., 2023) and DLinear (Zeng et al., 2023).

**Implementation Details.** All experiments are implemented in PyTorch (Paszke et al., 2019) and conducted on a single NVIDIA RTX 3090 24GB GPU. We use the Adam optimizer (Kingma & Ba, 2015) with a learning rate selected from $\{1e\text{-}3, 3e\text{-}3, 5e\text{-}4\}$. The hidden dim for MLP backbone $D$ is set to 512. For additional details on hyperparameters and settings, please refer to the Appendix A.3.

### 4.2 EXPERIMENTS RESULTS

#### 4.2.1 LONG-TERM FORECASTING

Table 1: Long-term forecasting results. The look-back length $T$ is fixed at 96. All results are averaged across four different forecasting horizons $S \in \{96, 192, 336, 720\}$. The best results are highlighted in **bold**, while the second-best results are underlined. Count row counts the number of times each model ranks in the top 2. See Table 7 in Appendix C.1 for the full results.

| Model | GTR (Ours) | | RAFT (2025) | | S-Mamba (2025b) | | TQNet (2025a) | | TimeXer (2024d) | | CycleNet (2024) | | SOFTS (2024) | | TimeMixer (2024c) | | iTransformer (2024b) | | PatchTST (2023) | | DLinear (2023) | |
|---|---|---|---|---|---|---|---|---|---|---|---|---|---|---|---|---|---|---|---|---|---|---|
| Metric | MSE | MAE | MSE | MAE | MSE | MAE | MSE | MAE | MSE | MAE | MSE | MAE | MSE | MAE | MSE | MAE | MSE | MAE | MSE | MAE | MSE | MAE |
| ETTh1 | 0.439 | 0.434 | **0.428** | **0.433** | 0.455 | 0.450 | 0.441 | 0.434 | 0.437 | 0.437 | 0.457 | 0.441 | 0.449 | 0.443 | 0.447 | 0.440 | 0.454 | 0.448 | 0.469 | 0.455 | 0.456 | 0.452 |
| ETTh2 | 0.372 | 0.400 | 0.382 | 0.410 | 0.381 | 0.405 | 0.378 | 0.402 | 0.368 | 0.396 | 0.388 | 0.409 | 0.373 | 0.400 | **0.365** | **0.395** | 0.383 | 0.407 | 0.387 | 0.407 | 0.559 | 0.515 |
| ETTm1 | **0.367** | **0.389** | 0.381 | 0.400 | 0.398 | 0.405 | 0.377 | 0.393 | 0.382 | 0.397 | 0.379 | 0.396 | 0.393 | 0.402 | 0.381 | 0.396 | 0.387 | 0.410 | 0.387 | 0.400 | 0.403 | 0.407 |
| ETTm2 | 0.268 | 0.315 | 0.281 | 0.330 | 0.288 | 0.332 | 0.277 | 0.323 | 0.274 | 0.322 | **0.266** | **0.314** | 0.287 | 0.330 | 0.275 | 0.323 | 0.288 | 0.332 | 0.281 | 0.326 | 0.350 | 0.401 |
| Electricity | 0.166 | 0.260 | 0.175 | 0.272 | 0.170 | 0.265 | **0.164** | **0.259** | 0.171 | 0.270 | 0.168 | 0.259 | 0.174 | 0.264 | 0.182 | 0.273 | 0.178 | 0.270 | 0.205 | 0.290 | 0.212 | 0.300 |
| Solar | **0.194** | **0.245** | 0.301 | 0.303 | 0.240 | 0.273 | 0.198 | 0.256 | 0.237 | 0.302 | 0.210 | 0.261 | 0.229 | 0.256 | 0.216 | 0.280 | 0.233 | 0.262 | 0.270 | 0.307 | 0.330 | 0.401 |
| Traffic | 0.470 | 0.280 | 0.414 | 0.284 | 0.414 | 0.276 | 0.445 | 0.276 | 0.466 | 0.287 | 0.472 | 0.301 | **0.409** | **0.267** | 0.485 | 0.298 | 0.428 | 0.282 | 0.481 | 0.300 | 0.625 | 0.383 |
| Weather | **0.239** | **0.268** | 0.270 | 0.309 | 0.251 | 0.276 | 0.242 | 0.269 | 0.241 | 0.271 | 0.243 | 0.271 | 0.255 | 0.278 | 0.240 | 0.272 | 0.258 | 0.278 | 0.259 | 0.273 | 0.265 | 0.317 |
| Count | **5** | **5** | 2 | 1 | 0 | 1 | 3 | 4 | 2 | 1 | 1 | 2 | 1 | 1 | 2 | 1 | 0 | 0 | 0 | 0 | 0 | 0 |

**Results.** Table 1 shows GTR outperforms other models in long-term forecasting across various datasets. Across 16 prediction tasks, GTR achieved top-2 performance in 10 of them. Due to the traffic dataset's strong spatiotemporal dependencies and temporal lag effects, GTR exhibits higher MSE than S-Mamba, SOFTS and iTransformer, as these models better capture critical inter-variable relationships. GTR is good at handling complex high-dimensional time series. For example, on the challenging Solar-Energy dataset, it exceeds the CycleNet by 8.2% in MSE and 6.5% in MAE.

### 4.2.2 SHORT-TERM FORECASTING

Table 2: Short-term forecasting results. The look-back length $T$ is fixed at 96. All results are averaged across four different forecasting horizons $S \in \{12, 24, 48, 96\}$. The best results are highlighted in **bold**, while the second-best results are underlined. Count row counts the number of times each model ranks in the top 2. See Table 7 in Appendix C.1 for the full results.

| Model | GTR (Ours) | | RAFT (2025) | | S-Mamba (2025b) | | TQNet (2025a) | | TimeXer (2024d) | | CycleNet (2024) | | SOFTS (2024) | | TimeMixer (2024c) | | iTransformer (2024b) | | PatchTST (2023) | | DLinear (2023) | |
|---|---|---|---|---|---|---|---|---|---|---|---|---|---|---|---|---|---|---|---|---|---|---|
| Metric | MSE | MAE | MSE | MAE | MSE | MAE | MSE | MAE | MSE | MAE | MSE | MAE | MSE | MAE | MSE | MAE | MSE | MAE | MSE | MAE | MSE | MAE |
| PEMS03 | **0.087** | **0.189** | 0.144 | 0.230 | 0.122 | 0.228 | 0.097 | 0.203 | 0.112 | 0.214 | 0.118 | 0.226 | 0.104 | 0.210 | 0.154 | 0.278 | 0.113 | 0.222 | 0.180 | 0.291 | 0.278 | 0.375 |
| PEMS04 | **0.087** | **0.189** | 0.104 | 0.210 | 0.103 | 0.211 | 0.091 | 0.197 | 0.105 | 0.209 | 0.119 | 0.232 | 0.102 | 0.208 | 0.156 | 0.282 | 0.111 | 0.221 | 0.195 | 0.307 | 0.295 | 0.388 |
| PEMS07 | 0.076 | 0.169 | 0.094 | 0.193 | 0.089 | 0.188 | **0.075** | 0.171 | 0.085 | 0.182 | 0.113 | 0.214 | 0.086 | 0.184 | 0.143 | 0.259 | 0.101 | 0.204 | 0.211 | 0.303 | 0.329 | 0.396 |
| PEMS08 | 0.142 | 0.222 | 0.151 | 0.234 | 0.148 | 0.224 | 0.142 | 0.230 | 0.175 | 0.250 | 0.150 | 0.246 | **0.138** | **0.220** | 0.253 | 0.336 | 0.150 | 0.226 | 0.280 | 0.321 | 0.379 | 0.416 |
| Count | 4 | 4 | 0 | 0 | 0 | 0 | 4 | 3 | 0 | 0 | 0 | 0 | 1 | 1 | 0 | 0 | 0 | 0 | 0 | 0 | 0 | 0 |

**Results.** Table 2 shows GTR outperforms state-of-the-art models across all metrics. Across 8 prediction tasks, GTR achieved top-2 performance in all of them. Compared to iTransformer, GTR reduces MSE by 18.7% and MAE by 12.1% on average across all datasets and horizons; compared to S-Mamba, it achieves 15.7% MSE and 9.6% MAE reductions, with the most significant gains observed in PEMS03, achieving 28.7% MSE and 20.6% MAE reduction.

### 4.3 ABLATION STUDIES AND ANALYSIS

**Effectiveness of GTR.** To evaluate this, we conducted comprehensive ablation studies across three highly periodic benchmark datasets. Table 3 systematically quantifies both the intrinsic contribution of GTR within our framework and its cross-model generalization capability when integrated into diverse state-of-the-art architectures.

Table 3: Ablation studies of the GTR technique. Left: Generalization performance of GTR technique on different models. Right: Ablation study revealing the effectiveness of the GTR technique. The look-back length is fixed at 96 for all the experiments.

| Model | | iTransformer (2024b) | | | | | | PatchTST (2023) | | | | | | DLinear (2023) | | | | | | MLP-Layer (Ours) | | | | | |
|---|---|---|---|---|---|---|---|---|---|---|---|---|---|---|---|---|---|---|---|---|---|---|---|---|---|---|
| Setup | | Original | | + GTR Tech. | | Improve ↑ | | Original | | + GTR Tech. | | Improve ↑ | | Original | | + GTR Tech. | | Improve ↑ | | Original | | + GTR Tech. | | Improve ↑ | |
| Metric | | MSE | MAE | MSE | MAE | MSE | MAE | MSE | MAE | MSE | MAE | MSE | MAE | MSE | MAE | MSE | MAE | MSE | MAE | MSE | MAE | MSE | MAE | MSE | MAE |
| Electricity 96 | | 0.150 | 0.241 | **0.136** | **0.231** | 10.2% | 4.3% | 0.164 | 0.254 | **0.135** | **0.233** | 21.4% | 9.0% | 0.195 | 0.277 | **0.141** | **0.240** | 38.2% | 15.4% | 0.165 | 0.252 | **0.134** | **0.229** | 23.1% | 10.0% |
| Electricity 192 | | 0.164 | 0.255 | **0.155** | **0.250** | 5.8% | 2.0% | 0.174 | 0.265 | **0.153** | **0.249** | 13.7% | 6.4% | 0.194 | 0.279 | **0.159** | **0.257** | 22.0% | 8.5% | 0.173 | 0.260 | **0.152** | **0.245** | 13.8% | 6.1% |
| Electricity 336 | | 0.178 | 0.272 | **0.166** | **0.263** | 7.2% | 3.4% | 0.193 | 0.285 | **0.172** | **0.270** | 12.2% | 5.5% | 0.207 | 0.295 | **0.173** | **0.274** | 19.6% | 7.6% | 0.190 | 0.277 | **0.171** | **0.264** | 11.1% | 4.9% |
| Electricity 720 | | 0.210 | 0.300 | **0.192** | **0.288** | 9.3% | 4.1% | 0.232 | 0.320 | **0.211** | **0.306** | 9.9% | 4.5% | 0.244 | 0.330 | **0.206** | **0.305** | 18.4% | 8.1% | 0.230 | 0.312 | **0.208** | **0.300** | 10.5% | 4.0% |
| Electricity Avg | | 0.175 | 0.267 | **0.162** | **0.258** | 8.0% | 3.5% | 0.191 | 0.281 | **0.167** | **0.264** | 14.3% | 6.4% | 0.210 | 0.295 | **0.169** | **0.269** | 24.2% | 9.6% | 0.190 | 0.276 | **0.166** | **0.260** | 14.4% | 6.1% |
| PEMS03 12 | | 0.106 | 0.219 | **0.067** | **0.172** | 58.2% | 27.3% | 0.072 | 0.179 | **0.058** | **0.160** | 24.1% | 11.8% | 0.105 | 0.221 | **0.072** | **0.180** | 45.8% | 22.8% | 0.071 | 0.177 | **0.057** | **0.156** | 24.5% | 13.4% |
| PEMS03 24 | | 0.090 | 0.199 | **0.074** | **0.178** | 21.6% | 11.8% | 0.102 | 0.213 | **0.073** | **0.178** | 39.7% | 19.6% | 0.183 | 0.299 | **0.104** | **0.214** | 75.9% | 39.7% | 0.103 | 0.212 | **0.070** | **0.172** | 47.1% | 23.2% |
| PEMS03 48 | | 0.199 | 0.304 | **0.113** | **0.215** | 76.1% | 41.3% | 0.155 | 0.263 | **0.101** | **0.207** | 53.4% | 16.0% | 0.315 | 0.407 | **0.155** | **0.258** | 103.2% | 57.7% | 0.158 | 0.265 | **0.094** | **0.198** | 68.1% | 33.8% |
| PEMS03 96 | | 0.242 | 0.348 | **0.141** | **0.238** | 71.6% | 46.2% | 0.204 | 0.305 | **0.131** | **0.236** | 55.7% | 29.2% | 0.455 | 0.508 | **0.199** | **0.295** | 128.6% | 72.2% | 0.208 | 0.309 | **0.127** | **0.228** | 63.7% | 27.5% |
| PEMS03 Avg | | 0.159 | 0.268 | **0.098** | **0.200** | 62.2% | 34.0% | 0.133 | 0.240 | **0.090** | **0.195** | 47.7% | 23.0% | 0.265 | 0.359 | **0.132** | **0.236** | 100.7% | 52.1% | 0.135 | 0.241 | **0.087** | **0.189** | 55.1% | 27.5% |
| PEMS04 12 | | 0.081 | 0.189 | **0.067** | **0.168** | 20.8% | 12.5% | 0.087 | 0.195 | **0.069** | **0.171** | 26.0% | 14.0% | 0.114 | 0.228 | **0.083** | **0.192** | 37.3% | 18.7% | 0.086 | 0.193 | **0.065** | **0.164** | 32.3% | 17.7% |
| PEMS04 24 | | 0.097 | 0.207 | **0.078** | **0.181** | 24.3% | 14.3% | 0.119 | 0.231 | **0.081** | **0.187** | 46.9% | 19.1% | 0.187 | 0.298 | **0.113** | **0.225** | 65.4% | 32.4% | 0.119 | 0.229 | **0.075** | **0.177** | 58.6% | 29.3% |
| PEMS04 48 | | 0.128 | 0.241 | **0.092** | **0.198** | 39.1% | 21.7% | 0.172 | 0.279 | **0.101** | **0.211** | 70.2% | 32.3% | 0.319 | 0.402 | **0.157** | **0.226** | 103.1% | 77.8% | 0.174 | 0.283 | **0.093** | **0.197** | 87.1% | 43.6% |
| PEMS04 96 | | 0.176 | 0.284 | **0.113** | **0.221** | 55.7% | 28.5% | 0.221 | 0.323 | **0.136** | **0.249** | 62.5% | 29.7% | 0.424 | 0.481 | **0.193** | **0.298** | 119.6% | 61.4% | 0.224 | 0.328 | **0.114** | **0.219** | 96.5% | 49.8% |
| PEMS04 Avg | | 0.120 | 0.230 | **0.087** | **0.192** | 37.9% | 19.8% | 0.150 | 0.257 | **0.096** | **0.204** | 56.2% | 30.0% | 0.261 | 0.352 | **0.136** | **0.235** | 91.9% | 49.8% | 0.151 | 0.258 | **0.087** | **0.189** | 73.5% | 36.5% |

Key findings are twofold. First, GTR significantly enhances our MLP backbone, reducing average MSE by 14.4% (Electricity), 55.1% (PEMS03), and 73.5% (PEMS04). Second, GTR consistently improves all evaluated models across all datasets and prediction horizons, with particularly pronounced gains in traffic forecasting. For instance, on PEMS04, GTR reduces MSE by 91.9% for DLinear and 56.2% for PatchTST, underscoring its robustness in complex temporal contexts. While the GTR module itself is designed with a channel-independent architecture, when integrated with the iTransformer, which is specifically built to capture relationships between different variables, the GTR module still delivered significant improvements, reducing MSE by 62.2% on PEMS03 and 37.9% on PEMS04. These improvements are achieved through trivial integration during training, demonstrating GTR's capability with diverse architectures.

**Influence of look-back length.** Figure 3 demonstrates GTR's performance across varying look-back window lengths on four benchmark datasets. Two critical observations emerge: First, GTR exhibits consistent performance improvement as the look-back length increases, demonstrating effective utilization of additional temporal context. More significantly, GTR substantially outperforms all baseline methods across all window lengths, with the largest gains occurring at the shortest input horizons. Notably, on Electricity and Solar-Energy datasets, baseline models experience exponential error growth as window length decreases, while GTR maintains remarkable stability with only

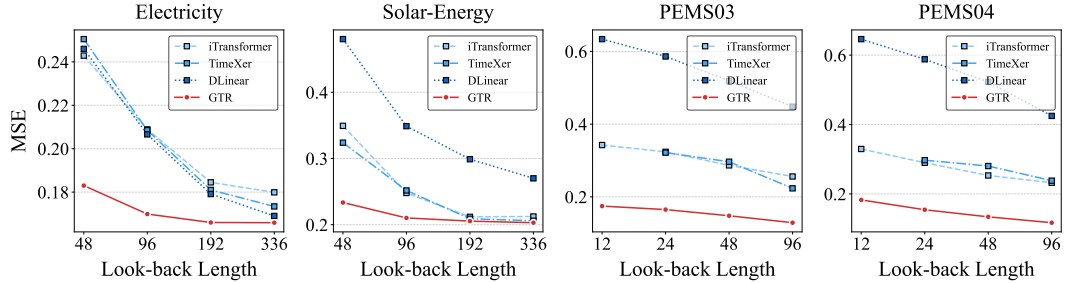

Figure 3: Influence of look-back length. Forecasting horizons are fixed at 336 for Electricity and Solar-Energy, and 96 for PEMS03 and PEMS04. GTR consistently outperforms other models across varying look-back lengths, particularly at the shortest window length.

marginal performance degradation. This resilience in ultra-short input scenarios directly validates our core innovation—by establishing temporal continuity beyond the immediate observation window, GTR successfully leverages global periodic information that remains inaccessible to baseline methods when cycle lengths exceed the input horizon. The consistent superiority across all window lengths confirms that GTR effectively bridges local pattern extraction with global temporal structure, making it particularly valuable for practical applications where historical data may be severely limited.

**Model Efficiency.** The proposed GTR technique operates as an efficient plug-and-play module with minimal computational overhead. As shown in Table 4, the GTR technique alone consumes merely 40.1K parameters and 4.50M MACs. When integrated with a pure MLP backbone for complete forecasting capability, the combined system maintains remarkable efficiency with just 0.98M parameters and 306.91M MACs, representing only 19.0% of iTransformer's parameter count while achieving comparable or superior prediction accuracy as demonstrated in our main results. With a training time of 22.3 seconds per epoch, the combined system is surpassed only by DLinear (18.1s), highlighting its strong efficiency while maintaining superior modeling capability. This makes GTR Technique particularly suitable for resource-constrained forecasting applications while maintaining the capacity to capture both short-term patterns and long-range dependencies.

Table 4: Efficiency comparison between GTR and other models on the Electricity dataset with look-back length $T = 96$ and forecast horizon $S = 720$. Training Time denotes the average time required per epoch for the model.

| Model | Parameters | MACs | Training Time(s) |
|---|---|---|---|
| Informer (2021) | 12.53M | 3.97G | 70.1 |
| Autoformer (2021) | 12.22M | 4.41G | 107.7 |
| FEDformer (2022a) | 17.98M | 4.41G | 238.7 |
| DLinear (2023) | 139.6K | 44.91M | 18.1 |
| PatchTST (2023) | 10.74M | 25.87G | 129.5 |
| iTransformer (2024b) | 5.15M | 1.65G | 35.1 |
| GTR Tech. | 40.1K | 4.50M | - |
| + MLP-Layer | 0.98M | 306.91M | 22.3 |

**Correlation Alignment.** To further investigate how the GTR module models global temporal dependencies, we conducted a visualization experiment using the Pearson correlation coefficient to analyze multivariate correlations. As shown in Figure 4, we selected test sequences from 4 different datasets. The visualization compares input and GTR-enhanced output against the ground-truth correlation computed across the entire dataset. The results clearly demonstrate that, after applying the GTR module, the learned multivariate correlations align significantly more closely with the global temporal structure. This improvement is particularly meaningful in real-world scenarios, where time series may be distorted by non-stationary disturbances. The GTR module enhances the model's robustness to such perturbations, enabling more reliable and structure-preserving representations even under imperfect input conditions.

## 5 LIMITATIONS AND FUTURE WORK

While our proposed GTR module offers a flexible, plug-and-play solution for enhancing long-term time series forecasting through explicit periodic pattern modeling, it exhibits several inherent limitations stemming from its architectural assumptions and computational design.

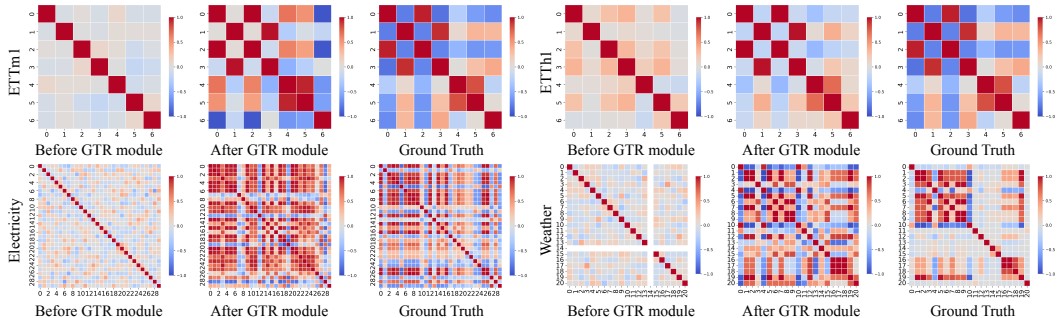

Figure 4: Visualization for multivariate correlation analysis on 4 datasets. The visualization is implemented based on the Pearson Correlation Coefficient. The Ground Truth denotes the correlation computed across the entire dataset. The two columns on the left denote the correlation between the variables before and after the GTR module, respectively. It shows that the model drives the learned multivariate correlations closer to the global correlation structures through the GTR module.

**Fixed Cycle-Length Assumption.** GTR assumes a single, fixed cycle length for the entire input sequence. This assumption may result in suboptimal performance on datasets with time-varying periodicity, such as physiological signals whose dominant frequencies shift over time. Moreover, GTR enforces a shared cycle length across all input channels, which is ill-suited for multivariate time series in which different variables may exhibit heterogeneous periodic patterns. Although this limitation could be mitigated through channel-specific modeling or preprocessing, such solutions introduce additional architectural complexity or data-handling overhead. GTR further assumes that time series simultaneously exhibit both local and global periodic structures. Consequently, it may underperform on time series characterized by a single dominant periodicity.

**Challenges in Modeling Long-Range Cycles:** Although GTR can theoretically accommodate long cycles by adjusting the cycle-length parameter, its practical applicability is often constrained by data scarcity. Accurately learning yearly or multi-year periodic patterns typically requires decades of continuous, high-quality historical data—a condition rarely satisfied in real-world settings. Moreover, when the base cycle length $P$ becomes large, the width of the 2D convolution kernel increases linearly with $P$. Consequently, the convolution operation incurs a computational complexity of $O(NTP)$, where $T$ denotes the sequence length. For large-scale time series and ultra-long cycles, such complexity becomes computationally prohibitive. Additionally, the number of kernel parameters also scales linearly with $P$, increasing memory overhead and exacerbating the risk of overfitting in data-scarce scenarios. Future work may investigate hierarchical or memory-augmented mechanisms to better extrapolate long-range periodic structures from limited observations.

**Scalability with Long Input Sequences.** GTR preserves the input sequence length and employs a linear projection layer with size $T \times T$. This results in a computational complexity of $O(NT^2)$ for the linear operation. When $T$ is large, the quadratic complexity becomes prohibitive, as the number of operations scales with the square of the sequence length. Although the GTR module is designed for short look-back windows, it becomes inefficient for long sequences.

These limitations highlight specific scenarios where the GTR module may not be the optimal solution. They also motivate future research into developing adaptive cycle modeling, handling cross-channel periodic heterogeneity, and improving the efficient learning of extreme long-horizon periodicity.

## 6  CONCLUSION

In this work, we address the challenge that existing multivariate time series forecasting (MTSF) models are limited by their historical context, which prevents them from effectively capturing global periodic patterns. To overcome this, we introduce the Global Temporal Retriever (GTR), a lightweight and plug-and-play module that allows any forecasting model to access and utilize temporal information from the entire cycle. Extensive experiments on six real-world datasets demonstrate that GTR consistently delivers state-of-the-art performance in both short-term and long-term forecasting scenarios, highlighting GTR as an efficient and general solution for MTSF tasks.

## 7 ETHICS STATEMENT

Our work only focuses on the time series forecasting problem, so there is no potential ethical risk.

## 8 REPRODUCIBILITY STATEMENT

In the main text, we have strictly formalized the model architecture with equations. All the implementation details are included in the Appendix, including dataset descriptions, metrics, model, and experiment configurations. The code has been made public.

## 9 ACKNOWLEDGMENTS

This work was supported in part by the National Key R&D Program of China (Grant No. 2023YFF0725001), in part by the National Natural Science Foundation of China (Grant No. 92370204), in part by the Guangdong Basic and Applied Basic Research Foundation (Grant No. 2023B1515120057), in part by the Key-Area Special Project of Guangdong Provincial Ordinary Universities (Grant No. 2024ZDZX1007), and in part by the Red Bird MPhil Program at the Hong Kong University of Science and Technology (Guangzhou).

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

## A   IMPLEMENTATION DETAILS

### A.1   DATASETS DETAILS

We conduct extensive experiments on five widely-used time series datasets for long-term forecasting and PEMS datasets for short-term forecasting. We report the statistics in Table 5. Detailed descriptions of these datasets are as follows:

(1) **ETT** (Electricity Transformer Temperature) dataset (Zhou et al., 2021) encompasses temperature and power load data from electricity transformers in two regions of China, spanning from 2016 to 2018. This dataset has two granularity levels: ETTh (hourly) and ETTm (15 minutes).

(2) **Weather** dataset (Wu et al., 2022) captures 21 distinct meteorological indicators in Germany, meticulously recorded at 10-minute intervals throughout 2020. Key indicators in this dataset include air temperature, visibility, among others, offering a comprehensive view of the weather dynamics.

(3) **Electricity** dataset (Wu et al., 2022) features hourly electricity consumption records in kilowatt-hours (kWh) for 321 clients. Sourced from the UCL Machine Learning Repository, this dataset covers the period from 2012 to 2014, providing valuable insights into consumer electricity usage patterns.

(4) **Traffic** dataset (Wu et al., 2022) includes data on hourly road occupancy rates, gathered by 862 detectors across the freeways of the San Francisco Bay area. This dataset, covering the years 2015 to 2016, offers a detailed snapshot of traffic flow and congestion.

(5) **Solar-Energy** dataset (Lin et al., 2024) contains solar power production data recorded every 10 minutes throughout 2006 from 137 photovoltaic (PV) plants in Alabama.

(6) **PEMS** dataset (Liu et al., 2022a) comprises four public traffic network datasets (PEMS03, PEMS04, PEMS07, and PEMS08), constructed from the Caltrans Performance Measurement System (PeMS) across four districts in California. The data is aggregated into 5-minute intervals, resulting in 12 data points per hour and 288 data points per day.

Table 5: Dataset detailed descriptions. "Dataset Size" denotes the total number of time points in (Train, Validation, Test) split respectively. "Prediction Length" denotes the future time points to be predicted. "Frequency" denotes the sampling interval of time points.

| Tasks | Dataset | Dim | Prediction Length | Dataset Size | Frequency |
|---|---|---|---|---|---|
| | ETTm1 | 7 | $\{96, 192, 336, 720\}$ | $(34465, 11521, 11521)$ | 15 min |
| | ETTm2 | 7 | $\{96, 192, 336, 720\}$ | $(34465, 11521, 11521)$ | 15 min |
| | ETTh1 | 7 | $\{96, 192, 336, 720\}$ | $(8545, 2881, 2881)$ | 15 min |
| Long-term | ETTh2 | 7 | $\{96, 192, 336, 720\}$ | $(8545, 2881, 2881)$ | 15 min |
| Forecasting | Weather | 21 | $\{96, 192, 336, 720\}$ | $(36792, 5271, 10540)$ | 10 min |
| | Electricity | 321 | $\{96, 192, 336, 720\}$ | $(18317, 2633, 5261)$ | 1 hour |
| | Traffic | 862 | $\{96, 192, 336, 720\}$ | $(12185, 1757, 3509)$ | 1 hour |
| | Solar-Energy | 137 | $\{96, 192, 336, 720\}$ | $(36601, 5161, 10417)$ | 10 min |
| | PEMS03 | 358 | $\{12, 24, 48, 96\}$ | $(15617, 5135, 5135)$ | 5 min |
| Short-term | PEMS04 | 307 | $\{12, 24, 48, 96\}$ | $(10172, 3375, 3375)$ | 5 min |
| Forecasting | PEMS07 | 883 | $\{12, 24, 48, 96\}$ | $(16911, 5622, 5622)$ | 5 min |
| | PEMS08 | 170 | $\{12, 24, 48, 96\}$ | $(10690, 3548, 265)$ | 5 min |

## A.2 METRIC DETAILS

Regarding metrics, we utilize the mean square error (MSE) and mean absolute error (MAE) for long-term forecasting. The calculations of these metrics are:

$$MSE = \frac{1}{T} \sum_{i=1}^{T} (\hat{y}_i - y_i)^2 \,, \qquad MAE = \frac{1}{T} \sum_{i=1}^{T} |\hat{y}_i - y_i|$$

## A.3 EXPERIMENT DETAILS

All experiments are implemented in PyTorch (Paszke et al., 2019) and conducted on a single NVIDIA RTX 3090 24GB GPU. We use the Adam optimizer (Kingma & Ba, 2015) with a learning rate selected from {1e-3, 3e-3, 5e-4}. The hidden dim for MLP backbone is set to 512. The dominant high-frequency period length $P$ is set to 24 for all datasets. Table 6 provides detailed hyperparameter settings for each dataset. During data preprocessing, we set the timestamp ($t_0$) of the initial observation in the entire dataset to 0 as the reference point. In main experiment, for all baseline methods, we adopted the hyperparameter settings from their respective original publications. For the ablation experiment of the GTR module in Table 3, the baseline methods utilized their original hyperparameter settings as specified in their respective publications for an input length of $T$=96. The GTR module was then directly integrated, and the entire model was trained end-to-end without any further hyperparameter optimization.

Table 6: Hyperparameter settings for different datasets. "lr" denotes the learning rate. "Cycle_len" denotes the length of global cycle.

| Tasks | Dataset | Cycle_len ($L$) | lr | Batchsize | Epochs | Use_revin |
|---|---|---|---|---|---|---|
| Long-term Forecasting | ETTm1 | 96 | 1e-3 | 256 | 30 | $True$ |
| | ETTm2 | 96 | 1e-3 | 256 | 30 | $True$ |
| | ETTh1 | 24 | 1e-3 | 256 | 30 | $True$ |
| | ETTh2 | 24 | 1e-3 | 256 | 30 | $True$ |
| | Weather | 144 | 1e-3 | 64 | 30 | $True$ |
| | Traffic | 168 | 3e-3 | 16 | 30 | $True$ |
| | Electricity | 168 | 3e-3 | 32 | 30 | $True$ |
| | Solar-Energy | 144 | 3e-3 | 64 | 30 | $False$ |
| Short-term Forecasting | PEMS03 | 288 | 3e-3 | 32 | 30 | $False$ |
| | PEMS04 | 288 | 3e-3 | 32 | 30 | $False$ |
| | PEMS07 | 288 | 3e-3 | 32 | 30 | $False$ |
| | PEMS08 | 288 | 3e-3 | 32 | 30 | $True$ |

The hyperparameter $L$, which denotes the global cycle length of the entire dataset, is established via two primary methods. The first relies on *domain-specific knowledge* of intrinsic periodic patterns and the *sampling intervals* of data, as suggested by prior work (Wu et al., 2022; Lin et al., 2024). The selected $L$ values are summarized in Table 6.

Alternatively, $L$ can be estimated using computational techniques, most notably the autocorrelation function (ACF) (Madsen, 2007). The ACF measures the linear dependence between a time series and a lagged version of itself. For a discrete time series $y_t$ of length $T$ with sample mean $\bar{y}$, the sample autocorrelation $r_k$ at lag $k$ is formally defined as the ratio of the sample autocovariance $\gamma_k$ to the sample variance $\gamma_0$:

$$r_k = \frac{\gamma_k}{\gamma_0} = \frac{\sum_{t=1}^{T-k} (y_t - \bar{y})(y_{t+k} - \bar{y})}{\sum_{t=1}^{T} (y_t - \bar{y})^2}$$

In this context, a significant peak in the ACF plot at a specific lag $k$ suggests a strong periodic component, justifying the selection of $L = k$.

# B  MORE DETAILS OF GTR

---

**Algorithm 1** Overall Pseudocode of GTR

---

**Require:** Historical look-back input $\boldsymbol{X} \in \mathbb{R}^{T \times N}$, starting absolute position $t_0$.
**Ensure:** Forecasting output $\boldsymbol{Y} \in \mathbb{R}^{S \times N}$
 1: Initialize **learnable** parameters $\boldsymbol{Q} \in \mathbb{R}^{L \times N} \leftarrow 0$
 2: **if** Instance Normalization is applied **then**
 3:    $\mu, \sigma \leftarrow \text{Mean}(\boldsymbol{X}), \text{STD}(\boldsymbol{X})$
 4:    $\boldsymbol{X} \leftarrow \frac{\boldsymbol{X} - \mu}{\sqrt{\sigma^2 + \epsilon}}$
 5: **end if**
 6: *// GTR part.*
 7: $\boldsymbol{i} \in \mathbb{N}_0^T \leftarrow \big[(t_0 \bmod L) + \tau\big] \bmod L \quad \text{for} \quad \tau = 0, 1, \ldots, T-1.$
 8: $\bar{\boldsymbol{Q}} \in \mathbb{R}^{T \times N} \leftarrow \text{Linear}(\boldsymbol{Q}[\boldsymbol{i}, :])$
 9: $\boldsymbol{F} \in \mathbb{R}^{2 \times T \times N} \leftarrow \begin{bmatrix} \boldsymbol{X} \\ \bar{\boldsymbol{Q}} \end{bmatrix}$
10: $\boldsymbol{H} \in \mathbb{R}^{T \times N} \leftarrow Conv2D(\boldsymbol{F})$
11: $\boldsymbol{Z} \in \mathbb{R}^{T \times N} \leftarrow \boldsymbol{H} + \boldsymbol{X}$
12: *// Host model part.*
13: $\boldsymbol{Z} \in \mathbb{R}^{D \times N} \leftarrow \text{Linear}(\boldsymbol{Z})$
14: $\boldsymbol{Z} \in \mathbb{R}^{D \times N} \leftarrow \text{Multi-Layer Perceptron}(\boldsymbol{Z}) + \boldsymbol{Z}$
15: $\bar{\boldsymbol{Y}} \in \mathbb{R}^{C \times H} \leftarrow \text{Linear}(\text{Dropout}(\boldsymbol{Z}))$
16: **if** Instance Normalization is applied **then**
17:    $\boldsymbol{Y} \leftarrow \bar{\boldsymbol{Y}} \times \sqrt{\sigma^2 + \epsilon} + \mu$
18: **end if**

---

The GTR framework processes temporal sequences through a structured pipeline that integrates cyclic pattern modeling with residual learning (Algorithm 1). Initially, optional instance normalization standardizes the input $\boldsymbol{X} \in \mathbb{R}^{T \times N}$ using per-sequence statistics $\mu$ and $\sigma$. The core innovation lies in the cyclic indexing mechanism: a learnable embedding matrix $\boldsymbol{Q} \in \mathbb{R}^{L \times N}$ is dynamically indexed via $\boldsymbol{i} = \big[(t_0 \bmod L) + \tau\big] \bmod L$ for $\tau = 0, \ldots, T-1$, generating position-aware queries $\bar{\boldsymbol{Q}}$ through a linear transformation. This indexed query is concatenated with the normalized input along the channel dimension to form $\boldsymbol{F} \in \mathbb{R}^{2 \times T \times N}$, which is processed by a 2D convolutional layer to extract spatiotemporal features $\boldsymbol{H}$. A residual connection $\boldsymbol{Z} = \boldsymbol{H} + \boldsymbol{X}$ preserves input integrity while enhancing representational capacity. The resulting features $\boldsymbol{Z}$ are then projected to the host model's input dimension, enabling seamless transition to downstream forecasting components.

Critically, GTR is designed as a plug-and-play module that requires *no architectural modifications* to the host model. As delineated in the pseudocode, the separation between the *GTR part* and *Host model part* ensures compatibility with any forecasting architecture (e.g., Transformers, RNNs, or MLPs). The host model receives $\boldsymbol{Z}$ as a direct replacement for its original input, eliminating the need for re-engineering existing layers or attention mechanisms. During training, all components—including $\boldsymbol{Q}$, the convolutional layer, and host model parameters—are jointly optimized end-to-end, allowing GTR to adaptively refine cyclic representations while the host model leverages these enriched features. This integration strategy significantly boosts forecasting accuracy by injecting explicit periodicity awareness without compromising the host model's inductive biases or increasing inference complexity, as evidenced by our empirical results across diverse benchmarks.

# C  MORE EXPERIMENT RESULTS

## C.1  FULL FORECASTING RESULTS

The full multivariate forecasting results are provided in this section due to the space limitation of the main text. We extensively evaluate competitive counterparts on challenging forecasting tasks. Table 7 contains the detailed results of all prediction lengths of the 12 well-acknowledged forecasting benchmarks. GTR achieves comprehensive state-of-the-art in real-world forecasting applications.

Table 7: Full multivariate time series forecasting results for all prediction horizons. The look-back length $T$ is fixed at 96. The best results are highlighted in **bold**, while the second-best results are underlined. *Avg* means the average results from all four prediction lengths.

| Model | | GTR (Ours) | | RAFT (2025) | | S-Mamba (2025b) | | TQNet (2025a) | | TimeXer (2024d) | | CycleNet (2024) | | SOFTS (2024) | | TimeMixer (2024c) | | iTransformer (2024b) | | PatchTST (2023) | | DLinear (2023) | |
|---|---|---|---|---|---|---|---|---|---|---|---|---|---|---|---|---|---|---|---|---|---|---|---|
| Metric | | MSE | MAE | MSE | MAE | MSE | MAE | MSE | MAE | MSE | MAE | MSE | MAE | MSE | MAE | MSE | MAE | MSE | MAE | MSE | MAE | MSE | MAE |
| ETTh1 | 96 | **0.367** | **0.391** | 0.376 | 0.401 | 0.386 | 0.405 | 0.371 | 0.393 | 0.382 | 0.403 | 0.375 | 0.395 | 0.381 | 0.399 | 0.375 | 0.400 | 0.386 | 0.405 | 0.414 | 0.419 | 0.386 | 0.400 |
| | 192 | **0.420** | **0.421** | 0.417 | 0.424 | 0.443 | 0.437 | 0.428 | 0.426 | 0.429 | 0.435 | 0.436 | 0.428 | 0.435 | 0.431 | 0.429 | 0.421 | 0.441 | 0.436 | 0.460 | 0.445 | 0.437 | 0.432 |
| | 336 | 0.476 | 0.447 | **0.454** | **0.441** | 0.489 | 0.468 | 0.476 | 0.446 | 0.468 | 0.448 | 0.496 | 0.455 | 0.480 | 0.452 | 0.484 | 0.458 | 0.487 | 0.458 | 0.501 | 0.466 | 0.481 | 0.459 |
| | 720 | 0.493 | 0.476 | **0.465** | **0.465** | 0.502 | 0.489 | 0.487 | 0.470 | 0.469 | 0.461 | 0.520 | 0.484 | 0.499 | 0.488 | 0.498 | 0.482 | 0.503 | 0.491 | 0.500 | 0.488 | 0.519 | 0.516 |
| | Avg | 0.439 | 0.434 | **0.428** | **0.433** | 0.455 | 0.450 | 0.441 | 0.434 | 0.437 | 0.437 | 0.457 | 0.441 | 0.449 | 0.443 | 0.447 | 0.440 | 0.454 | 0.448 | 0.469 | 0.455 | 0.456 | 0.452 |
| ETTh2 | 96 | 0.290 | 0.342 | 0.289 | 0.342 | 0.296 | 0.348 | 0.295 | 0.343 | 0.286 | 0.338 | 0.298 | 0.344 | 0.297 | 0.347 | 0.289 | 0.341 | 0.297 | 0.349 | 0.302 | 0.348 | 0.333 | 0.387 |
| | 192 | 0.362 | 0.389 | 0.382 | 0.400 | 0.376 | 0.396 | 0.367 | 0.393 | 0.363 | 0.389 | 0.372 | 0.396 | 0.373 | 0.394 | 0.372 | 0.392 | 0.380 | 0.400 | 0.388 | 0.400 | 0.477 | 0.476 |
| | 336 | 0.414 | 0.429 | 0.424 | 0.439 | 0.424 | 0.431 | 0.417 | 0.427 | 0.414 | 0.423 | 0.431 | 0.439 | 0.410 | 0.426 | 0.386 | 0.414 | 0.428 | 0.432 | 0.426 | 0.433 | 0.594 | 0.541 |
| | 720 | 0.423 | 0.442 | 0.433 | 0.457 | 0.426 | 0.444 | 0.433 | 0.446 | 0.408 | 0.432 | 0.450 | 0.458 | 0.411 | 0.433 | 0.412 | 0.434 | 0.427 | 0.445 | 0.431 | 0.446 | 0.831 | 0.657 |
| | Avg | 0.372 | 0.400 | 0.382 | 0.410 | 0.381 | 0.405 | 0.378 | 0.402 | 0.368 | 0.396 | 0.388 | 0.409 | 0.373 | 0.400 | 0.365 | 0.395 | 0.383 | 0.407 | 0.387 | 0.407 | 0.559 | 0.515 |
| ETTm1 | 96 | 0.305 | 0.349 | 0.325 | 0.369 | 0.333 | 0.368 | 0.311 | 0.353 | 0.318 | 0.356 | 0.319 | 0.360 | 0.325 | 0.361 | 0.320 | 0.357 | 0.334 | 0.368 | 0.329 | 0.367 | 0.345 | 0.372 |
| | 192 | 0.349 | 0.375 | 0.364 | 0.388 | 0.376 | 0.390 | 0.356 | 0.378 | 0.362 | 0.383 | 0.360 | 0.381 | 0.375 | 0.389 | 0.361 | 0.381 | 0.377 | 0.391 | 0.367 | 0.385 | 0.380 | 0.389 |
| | 336 | 0.380 | 0.398 | 0.391 | 0.407 | 0.408 | 0.413 | 0.390 | 0.401 | 0.389 | 0.407 | 0.389 | 0.403 | 0.405 | 0.412 | 0.390 | 0.404 | 0.426 | 0.420 | 0.399 | 0.410 | 0.413 | 0.413 |
| | 720 | 0.435 | 0.436 | 0.444 | 0.438 | 0.475 | 0.448 | 0.452 | 0.440 | 0.452 | 0.441 | 0.447 | 0.441 | 0.466 | 0.447 | 0.454 | 0.441 | 0.491 | 0.459 | 0.454 | 0.439 | 0.474 | 0.453 |
| | Avg | 0.367 | 0.389 | 0.381 | 0.400 | 0.398 | 0.405 | 0.377 | 0.393 | 0.382 | 0.397 | 0.379 | 0.396 | 0.393 | 0.402 | 0.381 | 0.396 | 0.407 | 0.410 | 0.387 | 0.400 | 0.403 | 0.407 |
| ETTm2 | 96 | 0.168 | 0.249 | 0.176 | 0.264 | 0.179 | 0.263 | 0.173 | 0.256 | 0.171 | 0.256 | 0.163 | 0.246 | 0.180 | 0.261 | 0.175 | 0.258 | 0.180 | 0.264 | 0.175 | 0.259 | 0.193 | 0.292 |
| | 192 | 0.232 | 0.292 | 0.241 | 0.306 | 0.250 | 0.309 | 0.238 | 0.298 | 0.237 | 0.299 | 0.229 | 0.290 | 0.246 | 0.306 | 0.237 | 0.299 | 0.250 | 0.309 | 0.241 | 0.302 | 0.284 | 0.362 |
| | 336 | 0.287 | 0.330 | 0.302 | 0.345 | 0.312 | 0.349 | 0.301 | 0.340 | 0.296 | 0.338 | 0.284 | 0.327 | 0.319 | 0.352 | 0.298 | 0.340 | 0.311 | 0.348 | 0.305 | 0.343 | 0.369 | 0.427 |
| | 720 | 0.386 | 0.390 | 0.404 | 0.405 | 0.411 | 0.406 | 0.397 | 0.396 | 0.392 | 0.394 | 0.389 | 0.391 | 0.405 | 0.401 | 0.391 | 0.396 | 0.412 | 0.407 | 0.402 | 0.400 | 0.554 | 0.522 |
| | Avg | 0.268 | 0.315 | 0.281 | 0.330 | 0.288 | 0.332 | 0.277 | 0.323 | 0.274 | 0.322 | 0.266 | 0.314 | 0.287 | 0.330 | 0.275 | 0.323 | 0.288 | 0.332 | 0.281 | 0.326 | 0.350 | 0.401 |
| Electricity | 96 | 0.134 | 0.229 | 0.152 | 0.257 | 0.139 | 0.235 | 0.134 | 0.229 | 0.140 | 0.242 | 0.136 | 0.223 | 0.143 | 0.233 | 0.153 | 0.247 | 0.148 | 0.240 | 0.181 | 0.270 | 0.197 | 0.282 |
| | 192 | 0.152 | 0.245 | 0.156 | 0.257 | 0.159 | 0.255 | 0.154 | 0.247 | 0.157 | 0.256 | 0.152 | 0.244 | 0.158 | 0.248 | 0.166 | 0.256 | 0.162 | 0.253 | 0.188 | 0.274 | 0.196 | 0.285 |
| | 336 | 0.171 | 0.264 | 0.171 | 0.264 | 0.176 | 0.272 | 0.169 | 0.264 | 0.176 | 0.275 | 0.170 | 0.264 | 0.178 | 0.269 | 0.185 | 0.277 | 0.178 | 0.269 | 0.204 | 0.293 | 0.209 | 0.301 |
| | 720 | 0.208 | 0.300 | 0.221 | 0.300 | 0.204 | 0.298 | 0.201 | 0.294 | 0.211 | 0.306 | 0.212 | 0.299 | 0.218 | 0.305 | 0.225 | 0.310 | 0.225 | 0.317 | 0.246 | 0.324 | 0.245 | 0.333 |
| | Avg | 0.166 | 0.260 | 0.175 | 0.272 | 0.170 | 0.265 | 0.164 | 0.259 | 0.171 | 0.270 | 0.168 | 0.259 | 0.174 | 0.264 | 0.182 | 0.273 | 0.178 | 0.270 | 0.205 | 0.290 | 0.212 | 0.300 |
| Solar-Energy | 96 | 0.176 | 0.235 | 0.278 | 0.278 | 0.205 | 0.244 | 0.173 | 0.233 | 0.215 | 0.295 | 0.190 | 0.247 | 0.200 | 0.230 | 0.189 | 0.259 | 0.203 | 0.237 | 0.234 | 0.286 | 0.290 | 0.378 |
| | 192 | 0.193 | 0.244 | 0.297 | 0.297 | 0.237 | 0.270 | 0.199 | 0.257 | 0.236 | 0.301 | 0.210 | 0.266 | 0.229 | 0.253 | 0.222 | 0.283 | 0.233 | 0.261 | 0.267 | 0.310 | 0.320 | 0.398 |
| | 336 | 0.201 | 0.250 | 0.311 | 0.311 | 0.258 | 0.288 | 0.211 | 0.263 | 0.252 | 0.307 | 0.217 | 0.266 | 0.243 | 0.269 | 0.231 | 0.292 | 0.248 | 0.273 | 0.290 | 0.315 | 0.353 | 0.415 |
| | 720 | 0.205 | 0.251 | 0.318 | 0.325 | 0.260 | 0.288 | 0.209 | 0.270 | 0.244 | 0.305 | 0.223 | 0.266 | 0.245 | 0.272 | 0.223 | 0.285 | 0.249 | 0.275 | 0.289 | 0.317 | 0.356 | 0.413 |
| | Avg | 0.194 | 0.245 | 0.301 | 0.303 | 0.240 | 0.273 | 0.198 | 0.256 | 0.237 | 0.302 | 0.210 | 0.261 | 0.229 | 0.256 | 0.216 | 0.280 | 0.233 | 0.262 | 0.270 | 0.307 | 0.330 | 0.401 |
| Traffic | 96 | 0.440 | 0.263 | 0.388 | 0.265 | 0.382 | 0.261 | 0.413 | 0.261 | 0.428 | 0.271 | 0.458 | 0.296 | 0.376 | 0.251 | 0.462 | 0.285 | 0.395 | 0.268 | 0.462 | 0.290 | 0.650 | 0.396 |
| | 192 | 0.454 | 0.274 | 0.400 | 0.278 | 0.396 | 0.267 | 0.432 | 0.271 | 0.448 | 0.282 | 0.457 | 0.294 | 0.398 | 0.261 | 0.473 | 0.296 | 0.417 | 0.276 | 0.466 | 0.290 | 0.598 | 0.370 |
| | 336 | 0.472 | 0.282 | 0.411 | 0.287 | 0.417 | 0.276 | 0.450 | 0.277 | 0.473 | 0.289 | 0.470 | 0.299 | 0.415 | 0.269 | 0.498 | 0.296 | 0.433 | 0.283 | 0.482 | 0.300 | 0.605 | 0.373 |
| | 720 | 0.514 | 0.301 | 0.456 | 0.305 | 0.460 | 0.300 | 0.486 | 0.295 | 0.516 | 0.307 | 0.502 | 0.314 | 0.447 | 0.287 | 0.506 | 0.313 | 0.467 | 0.302 | 0.514 | 0.320 | 0.645 | 0.394 |
| | Avg | 0.470 | 0.280 | 0.414 | 0.284 | 0.414 | 0.276 | 0.445 | 0.276 | 0.466 | 0.287 | 0.472 | 0.301 | 0.409 | 0.267 | 0.485 | 0.298 | 0.428 | 0.282 | 0.481 | 0.300 | 0.625 | 0.383 |
| Weather | 96 | 0.154 | 0.200 | 0.185 | 0.241 | 0.165 | 0.210 | 0.157 | 0.200 | 0.157 | 0.205 | 0.158 | 0.203 | 0.166 | 0.208 | 0.163 | 0.209 | 0.174 | 0.214 | 0.177 | 0.210 | 0.196 | 0.255 |
| | 192 | 0.202 | 0.244 | 0.237 | 0.287 | 0.214 | 0.252 | 0.206 | 0.245 | 0.204 | 0.247 | 0.207 | 0.247 | 0.217 | 0.253 | 0.208 | 0.250 | 0.221 | 0.254 | 0.225 | 0.250 | 0.237 | 0.296 |
| | 336 | 0.259 | 0.287 | 0.290 | 0.327 | 0.274 | 0.297 | 0.262 | 0.287 | 0.261 | 0.290 | 0.262 | 0.289 | 0.282 | 0.300 | 0.251 | 0.287 | 0.278 | 0.296 | 0.278 | 0.290 | 0.283 | 0.335 |
| | 720 | 0.341 | 0.342 | 0.367 | 0.380 | 0.350 | 0.345 | 0.344 | 0.342 | 0.340 | 0.341 | 0.344 | 0.344 | 0.356 | 0.351 | 0.339 | 0.341 | 0.358 | 0.349 | 0.354 | 0.340 | 0.345 | 0.381 |
| | Avg | 0.239 | 0.268 | 0.270 | 0.309 | 0.251 | 0.276 | 0.242 | 0.269 | 0.241 | 0.271 | 0.243 | 0.271 | 0.255 | 0.278 | 0.240 | 0.272 | 0.258 | 0.278 | 0.259 | 0.273 | 0.265 | 0.317 |
| PEMS03 | 12 | 0.057 | 0.156 | 0.071 | 0.175 | 0.065 | 0.169 | 0.060 | 0.161 | 0.070 | 0.173 | 0.066 | 0.172 | 0.064 | 0.165 | 0.091 | 0.215 | 0.071 | 0.174 | 0.099 | 0.216 | 0.122 | 0.243 |
| | 24 | 0.070 | 0.172 | 0.099 | 0.201 | 0.087 | 0.196 | 0.077 | 0.182 | 0.092 | 0.194 | 0.089 | 0.201 | 0.083 | 0.188 | 0.115 | 0.242 | 0.093 | 0.201 | 0.142 | 0.259 | 0.201 | 0.317 |
| | 48 | 0.094 | 0.198 | 0.153 | 0.243 | 0.133 | 0.243 | 0.104 | 0.215 | 0.129 | 0.229 | 0.136 | 0.247 | 0.114 | 0.223 | 0.173 | 0.302 | 0.125 | 0.236 | 0.211 | 0.319 | 0.333 | 0.425 |
| | 96 | 0.127 | 0.228 | 0.253 | 0.301 | 0.201 | 0.305 | 0.148 | 0.253 | 0.157 | 0.261 | 0.182 | 0.282 | 0.156 | 0.264 | 0.238 | 0.355 | 0.164 | 0.275 | 0.269 | 0.370 | 0.457 | 0.515 |
| | Avg | 0.087 | 0.189 | 0.144 | 0.230 | 0.122 | 0.228 | 0.097 | 0.203 | 0.112 | 0.214 | 0.118 | 0.226 | 0.104 | 0.210 | 0.154 | 0.278 | 0.113 | 0.222 | 0.180 | 0.291 | 0.278 | 0.375 |
| PEMS04 | 12 | 0.065 | 0.164 | 0.077 | 0.179 | 0.076 | 0.180 | 0.067 | 0.166 | 0.074 | 0.178 | 0.078 | 0.186 | 0.074 | 0.176 | 0.103 | 0.228 | 0.078 | 0.183 | 0.105 | 0.224 | 0.148 | 0.272 |
| | 24 | 0.075 | 0.177 | 0.091 | 0.195 | 0.084 | 0.193 | 0.077 | 0.181 | 0.087 | 0.195 | 0.099 | 0.212 | 0.088 | 0.194 | 0.122 | 0.249 | 0.095 | 0.205 | 0.153 | 0.275 | 0.224 | 0.340 |
| | 48 | 0.093 | 0.197 | 0.111 | 0.218 | 0.115 | 0.224 | 0.097 | 0.206 | 0.110 | 0.214 | 0.133 | 0.248 | 0.110 | 0.219 | 0.167 | 0.298 | 0.120 | 0.233 | 0.229 | 0.339 | 0.355 | 0.437 |
| | 96 | 0.114 | 0.219 | 0.137 | 0.247 | 0.137 | 0.248 | 0.123 | 0.233 | 0.148 | 0.251 | 0.167 | 0.281 | 0.135 | 0.244 | 0.231 | 0.353 | 0.150 | 0.262 | 0.291 | 0.389 | 0.452 | 0.504 |
| | Avg | 0.087 | 0.189 | 0.104 | 0.210 | 0.103 | 0.211 | 0.091 | 0.197 | 0.105 | 0.209 | 0.119 | 0.232 | 0.102 | 0.208 | 0.156 | 0.282 | 0.111 | 0.221 | 0.195 | 0.307 | 0.295 | 0.388 |
| PEMS07 | 12 | 0.051 | 0.142 | 0.062 | 0.159 | 0.063 | 0.159 | 0.051 | 0.143 | 0.057 | 0.152 | 0.062 | 0.162 | 0.057 | 0.152 | 0.086 | 0.205 | 0.067 | 0.165 | 0.095 | 0.207 | 0.115 | 0.242 |
| | 24 | 0.063 | 0.156 | 0.078 | 0.178 | 0.081 | 0.183 | 0.063 | 0.159 | 0.079 | 0.179 | 0.086 | 0.192 | 0.073 | 0.173 | 0.111 | 0.235 | 0.088 | 0.190 | 0.150 | 0.262 | 0.210 | 0.329 |
| | 48 | 0.082 | 0.178 | 0.102 | 0.203 | 0.093 | 0.192 | 0.081 | 0.179 | 0.099 | 0.191 | 0.128 | 0.234 | 0.096 | 0.195 | 0.161 | 0.281 | 0.110 | 0.215 | 0.253 | 0.340 | 0.398 | 0.458 |
| | 96 | 0.110 | 0.202 | 0.135 | 0.231 | 0.117 | 0.217 | 0.103 | 0.203 | 0.107 | 0.205 | 0.176 | 0.268 | 0.120 | 0.218 | 0.215 | 0.315 | 0.139 | 0.245 | 0.346 | 0.404 | 0.594 | 0.553 |
| | Avg | 0.076 | 0.169 | 0.094 | 0.193 | 0.089 | 0.188 | 0.075 | 0.171 | 0.085 | 0.182 | 0.113 | 0.214 | 0.086 | 0.184 | 0.143 | 0.259 | 0.101 | 0.204 | 0.211 | 0.303 | 0.329 | 0.396 |
| PEMS08 | 12 | 0.071 | 0.168 | 0.078 | 0.181 | 0.076 | 0.178 | 0.071 | 0.170 | 0.075 | 0.176 | 0.082 | 0.185 | 0.074 | 0.171 | 0.089 | 0.201 | 0.079 | 0.182 | 0.168 | 0.232 | 0.154 | 0.276 |
| | 24 | 0.095 | 0.192 | 0.102 | 0.205 | 0.104 | 0.209 | 0.096 | 0.196 | 0.102 | 0.201 | 0.117 | 0.226 | 0.104 | 0.201 | 0.166 | 0.283 | 0.115 | 0.219 | 0.224 | 0.281 | 0.248 | 0.353 |
| | 48 | 0.148 | 0.234 | 0.148 | 0.244 | 0.167 | 0.228 | 0.149 | 0.244 | 0.158 | 0.248 | 0.169 | 0.268 | 0.164 | 0.253 | 0.270 | 0.369 | 0.186 | 0.235 | 0.321 | 0.354 | 0.440 | 0.470 |
| | 96 | 0.256 | 0.293 | 0.277 | 0.307 | 0.245 | 0.280 | 0.253 | 0.309 | 0.366 | 0.377 | 0.233 | 0.306 | 0.211 | 0.253 | 0.486 | 0.493 | 0.221 | 0.267 | 0.408 | 0.417 | 0.674 | 0.565 |
| | Avg | 0.142 | 0.222 | 0.151 | 0.234 | 0.148 | 0.224 | 0.142 | 0.230 | 0.175 | 0.250 | 0.150 | 0.246 | 0.138 | 0.220 | 0.253 | 0.336 | 0.150 | 0.226 | 0.280 | 0.321 | 0.379 | 0.416 |
| $1^{st}$ Count | | 32 | 36 | 5 | 2 | 1 | 0 | 7 | 2 | 2 | 3 | 4 | 5 | 5 | 8 | 4 | 2 | 0 | 1 | 0 | 0 | 0 | 0 |

## C.2 MORE ABLATION RESULTS

To further validate the architectural decisions of the Global Temporal Retriever (GTR) and assess the robustness of its components, we conducted a comprehensive ablation study focusing on two key aspects: the impact of Reversible Instance Normalization (RevIN) (Kim et al., 2021) and the design of the temporal pattern extraction module. The experimental results are summarized in Table 8.

### C.2.1 IMPACT OF REVERSIBLE INSTANCE NORMALIZATION

We first investigate the contribution of Reversible Instance Normalization (RevIN) to the model's performance. RevIN is designed to mitigate the distribution shift problem in time series forecasting by normalizing the input and denormalizing the output.

As observed in the left part of Table 8, the standard GTR (w. RevIN) significantly outperforms the version without RevIN (w/o RevIN) on the majority of datasets, particularly the ETT series and Weather datasets. For example, on the ETTh2 dataset (prediction horizon 720), removing RevIN

Table 8: Ablation results of RevIN and Model Variants. The best results in each comparison group are highlighted in **bold**.

| Model | | GTR w. RevIN | | GTR w/o. RevIN | | Original 2D Conv. | | Variant 1 Concat | | Variant 2 Inception | | Variant 3 1D Conv. | |
|---|---|---|---|---|---|---|---|---|---|---|---|---|---|
| Metric | | MSE | MAE | MSE | MAE | MSE | MAE | MSE | MAE | MSE | MAE | MSE | MAE |
| ETTh1 | 96 | **0.367** | **0.391** | 0.380 | 0.402 | **0.367** | **0.391** | 0.369 | 0.393 | 0.371 | 0.395 | 0.369 | 0.392 |
| | 192 | **0.420** | **0.421** | 0.433 | 0.437 | **0.420** | **0.421** | 0.423 | 0.423 | 0.423 | 0.422 | 0.427 | 0.423 |
| | 336 | **0.476** | **0.447** | 0.480 | 0.464 | 0.476 | **0.447** | **0.472** | 0.448 | 0.481 | 0.451 | 0.479 | 0.448 |
| | 720 | **0.493** | **0.476** | 0.540 | 0.529 | 0.493 | 0.476 | **0.490** | **0.473** | 0.507 | 0.476 | 0.513 | 0.487 |
| ETTh2 | 96 | **0.290** | 0.342 | 0.315 | 0.363 | **0.290** | **0.342** | 0.293 | 0.344 | 0.291 | **0.341** | 0.297 | 0.346 |
| | 192 | **0.362** | **0.389** | 0.422 | 0.430 | **0.362** | **0.389** | 0.375 | 0.394 | 0.369 | 0.391 | 0.372 | 0.393 |
| | 336 | **0.414** | **0.429** | 0.567 | 0.518 | **0.414** | 0.429 | 0.416 | **0.428** | 0.422 | 0.433 | 0.420 | 0.431 |
| | 720 | **0.423** | **0.442** | 0.872 | 0.649 | **0.423** | **0.442** | 0.452 | 0.455 | 0.424 | 0.443 | 0.424 | 0.442 |
| ETTm1 | 96 | **0.305** | **0.349** | 0.315 | 0.358 | **0.305** | **0.349** | 0.306 | 0.351 | 0.313 | 0.358 | 0.307 | 0.351 |
| | 192 | **0.349** | **0.375** | 0.356 | 0.389 | **0.349** | **0.375** | 0.350 | 0.376 | 0.352 | 0.377 | 0.355 | 0.378 |
| | 336 | 0.380 | **0.398** | **0.378** | 0.403 | 0.380 | 0.398 | **0.377** | **0.397** | 0.379 | 0.398 | 0.382 | 0.398 |
| | 720 | **0.435** | **0.436** | 0.442 | 0.451 | 0.435 | 0.436 | 0.434 | 0.435 | 0.442 | 0.436 | **0.433** | **0.434** |
| ETTm2 | 96 | **0.168** | **0.249** | 0.190 | 0.285 | 0.168 | **0.249** | 0.169 | 0.250 | **0.167** | 0.251 | 0.168 | 0.251 |
| | 192 | **0.232** | **0.292** | 0.285 | 0.355 | **0.232** | **0.292** | 0.237 | 0.295 | 0.235 | 0.295 | 0.233 | 0.293 |
| | 336 | **0.287** | **0.330** | 0.526 | 0.474 | **0.287** | **0.330** | 0.290 | 0.333 | 0.291 | 0.332 | 0.289 | 0.333 |
| | 720 | **0.386** | **0.390** | 0.798 | 0.610 | **0.386** | **0.390** | 0.390 | 0.392 | 0.390 | 0.391 | 0.389 | 0.392 |
| Electricity | 96 | 0.134 | **0.229** | **0.132** | 0.230 | **0.134** | 0.229 | 0.136 | **0.228** | 0.135 | 0.229 | 0.135 | 0.229 |
| | 192 | 0.152 | **0.245** | **0.151** | 0.248 | **0.152** | **0.245** | 0.154 | 0.247 | 0.153 | 0.246 | 0.154 | 0.248 |
| | 336 | 0.171 | **0.264** | **0.167** | 0.267 | 0.171 | 0.264 | 0.172 | **0.263** | **0.169** | 0.264 | 0.170 | 0.264 |
| | 720 | **0.208** | **0.300** | **0.208** | 0.305 | 0.208 | 0.300 | 0.209 | 0.300 | **0.207** | **0.298** | 0.209 | 0.300 |
| Solar | 96 | 0.181 | 0.237 | **0.176** | **0.235** | **0.176** | 0.235 | 0.182 | **0.234** | 0.180 | 0.237 | 0.181 | 0.237 |
| | 192 | 0.196 | 0.245 | **0.193** | **0.244** | 0.193 | 0.244 | **0.192** | 0.245 | 0.195 | **0.243** | 0.196 | 0.245 |
| | 336 | 0.210 | 0.259 | **0.201** | **0.250** | **0.201** | **0.250** | 0.207 | 0.256 | 0.205 | 0.252 | 0.210 | 0.259 |
| | 720 | 0.232 | 0.277 | **0.205** | **0.251** | **0.205** | **0.251** | 0.212 | 0.266 | 0.212 | 0.255 | 0.232 | 0.277 |
| Weather | 96 | **0.154** | **0.200** | 0.157 | 0.220 | **0.154** | **0.200** | 0.156 | 0.201 | 0.156 | 0.201 | 0.156 | 0.201 |
| | 192 | **0.202** | **0.244** | 0.208 | 0.269 | **0.202** | **0.244** | 0.203 | 0.245 | 0.203 | 0.245 | 0.205 | 0.246 |
| | 336 | **0.259** | **0.287** | 0.263 | 0.318 | **0.259** | **0.287** | 0.262 | 0.288 | 0.261 | 0.288 | 0.261 | 0.288 |
| | 720 | **0.341** | **0.342** | 0.357 | 0.382 | 0.341 | 0.342 | 0.343 | 0.343 | 0.343 | 0.342 | **0.340** | **0.341** |
| $1^{st}$ Count | | **20** | **23** | 9 | 4 | 19 | 18 | 4 | 6 | 3 | 3 | 2 | 2 |

results in a catastrophic increase in MSE from 0.423 to 0.872. This confirms that RevIN is crucial for handling data with significant non-stationarity and distribution shifts.

Notably, the Electricity and Solar datasets exhibit a deviation from this trend, where the exclusion of RevIN yields superior performance. This phenomenon can be attributed to the prevalence of continuous zero-value segments (e.g., zero power generation at night), which may bias the mean statistics estimated by RevIN. Despite these specific instances, we retain RevIN as the default configuration to ensure general robustness across diverse forecasting scenarios.

### C.2.2 ANALYSIS OF EXTRACTION VARIANTS

The core of the GTR module is the extraction of temporal patterns by fusing the local observation $\mathbf{x}_n \in \mathbb{R}^T$ and the retrieved global context $\mathbf{q}_n \in \mathbb{R}^T$. Our proposed method utilizes a lightweight 2D convolution. To verify the optimality of this design, we compare it against three distinct variants:

**Variant 1 (Point-wise Fusion):** This variant assesses *whether temporal convolution is necessary*. We *concatenate* $\mathbf{x}_n$ and $\mathbf{q}_n$ along the feature dimension and apply a linear projection point-wise at each time step. This removes the receptive field, relying solely on the immediate alignment of local and global values:

$$\mathbf{h}_n = \text{Linear}(\text{Concat}(\mathbf{x}_n, \mathbf{q}_n)) \tag{6}$$

**Variant 2 (Inception Module):** To test whether *multi-scale temporal patterns* enhance the effectiveness of the feature fusion process, we replace the original single-kernel convolution with an Inception block (Szegedy et al., 2015). The Inception operator applies several parallel convolutions

with different kernel widths and aggregates their outputs through a $1 \times 1$ fusion layer. Formally, we denote the entire multi-branch operation as:

$$\mathbf{h}_n = \text{Inception}(\boldsymbol{F}_n),$$

where $\boldsymbol{F}_n \in \mathbb{R}^{2 \times T}$ is the stacked local and global features (same as in Equation 3), and Inception$(\cdot)$ represents the multi-scale convolutional extraction followed by channel-wise fusion.

**Variant 3 (1D Convolution):** This variant treats the local observation and global context as two separate channels of a 1D sequence. It applies a 1D convolution over the time dimension:

$$\mathbf{h}_n = \text{Conv1d}_{\text{in}=2,\ \text{out}=1}(\boldsymbol{F}_n). \tag{7}$$

The right side of Table 8 presents the comparison among these variants. The results demonstrate that our *Original 2D Convolution* strategy yields the most consistent and superior performance, achieving the best MSE in 19 out of 40 cases and the best MAE in 18 cases.

- **Necessity of Convolution:** The inferior performance of Variant 1 (Concat) compared to the Original model (e.g., ETTh2 horizon 720 MSE 0.423 vs. 0.452) highlights the importance of the receptive field provided by convolution operations to capture temporal dependencies.
- **Efficiency vs. Complexity:** Surprisingly, the multi-scale Variant 2 (Inception) does not outperform the simpler single-kernel 2D convolution. While theoretically more powerful, the increased parameter count may lead to overfitting on shorter sequences, or it suggests that the dominant periodicity is well-captured by a single, optimally sized kernel.
- **Structure Bias:** The Original 2D Conv outperforms Variant 3 (1D Conv). We hypothesize that the 2D convolution kernel enforces a stronger structural prior for aligning the local and global rows compared to treating them as abstract channels, thereby learning a more robust fusion of the retrieved global information.

In conclusion, the ablation studies confirm that the GTR module with 2D convolution and RevIN represents the most effective and robust configuration for multivariate time series forecasting.

### C.2.3 CAPABILITY WITH INTER-CHANNEL MODELING

Despite its strong performance across various datasets, the channel-independent architecture of GTR presents a challenge in domains where data exhibits strong spatio-temporal dependencies. For instance, the Traffic dataset exhibits significant *inter-variable dependencies*, as road network flows are highly correlated. Models that effectively capture these relationships typically outperform channel-independent approaches on such datasets (Liu et al., 2024b; Wang et al., 2024d). To demonstrate GTR's adaptability and its ability to integrate with inter-channel modeling, we introduce a novel Global Token Aggregation (GTA) module.

The GTA (Global Token Aggregation) module is designed to distill the most salient inter-channel relationships into a single, comprehensive global representation. This is achieved by first transforming the input embedding tensor $\boldsymbol{Q} \in \mathbb{R}^{T \times N}$ — where $T$ is the sequence length and $N$ is the number of channels — into a global token $g$. This global token is computed via channel-wise weighted aggregation, where the weights are derived from a softmax over the channel dimension. The resulting global context is then broadcast and fused back with the original channels to guide each channel's prediction with unified global awareness.

Formally, given the input global representation $\bar{\boldsymbol{Q}} \in \mathbb{R}^{T \times N}$, the process proceeds as follows:

1. **Channel Aggregation**: Compute channel-wise attention weights using softmax along the channel dimension (dimension 1, assuming 0-indexed axes). Then, aggregate across channels to obtain a global token $g \in \mathbb{R}^{T \times 1}$:

$$\text{weight} = \text{Softmax}(\bar{\boldsymbol{Q}}) \in \mathbb{R}^{T \times N}$$

$$g = \sum_{n=1}^{N} \bar{\boldsymbol{Q}}_{:,n} \cdot \text{weight}_{:,n} \in \mathbb{R}^{T \times 1}$$

Here, $\bar{\boldsymbol{Q}}_{:,n}$ denotes the $n$-th channel across all time steps, and the weighted sum produces a single global token per time step.

2. **Re-broadcast and Fusion**: The global token $g$ is broadcast across all $N$ channels to form a global guidance tensor, which is then fused (e.g., via addition or concatenation — though your formula suggests replacement) with the original representation:

$$Q_{\text{fused}} = g \cdot \mathbf{1}_{1 \times N} \in \mathbb{R}^{T \times N}$$

where $\mathbf{1}_{1 \times N}$ is a row vector of ones, and broadcasting replicates $g$ across the channel dimension.

By injecting this globally aggregated token into each channel, the GTA module enables each channel's prediction to be informed by the collective dynamics of all channels, thereby capturing inter-variable dependencies in a parameter-efficient manner.

The integration of the GTA module significantly boosts GTR's performance on the challenging traffic dataset, as shown in Table 9. The aggregated token provides the necessary inter-channel context, resulting in a substantial reduction in both Mean Squared Error (MSE) across all prediction horizons.

Table 9: Performance Comparison on the Traffic Dataset.

| Setup | | GTR | | + GTA module | | Improvement (%)↑ | |
|---|---|---|---|---|---|---|---|
| $T$ | $S$ | MSE | MAE | MSE | MAE | MSE↓ | MAE↓ |
| 96 | 96 | 0.440 | **0.264** | **0.399** | 0.265 | 9.3% | -0.4% |
| 96 | 192 | 0.459 | 0.275 | **0.418** | **0.273** | 8.8% | 0.7% |
| 96 | 336 | 0.473 | 0.283 | **0.431** | **0.279** | 8.8% | 1.4% |
| 96 | 720 | 0.515 | 0.302 | **0.462** | **0.298** | 10.4% | 1.3% |

This remarkable improvement confirms that while GTR is designed as a channel-independent model, its core architecture is highly adaptable. When augmented with a mechanism to incorporate inter-variable dependencies, it can achieve state-of-the-art performance on challenging datasets where such relationships are critical.

## C.3 ERROR BAR ANALYSIS

To further evaluate the robustness of our method, we conducted a comprehensive error bar analysis by evaluating GTR across multiple random seeds on diverse benchmark datasets. For each dataset, we report the standard deviation of MSE and MAE over 5 independent runs with different initialization seeds. As shown in Table 10 and Figure 5, GTR exhibits remarkably low variance (mostly below 0.001) across all datasets and forecasting horizons. This strongly indicates the robustness of GTR.

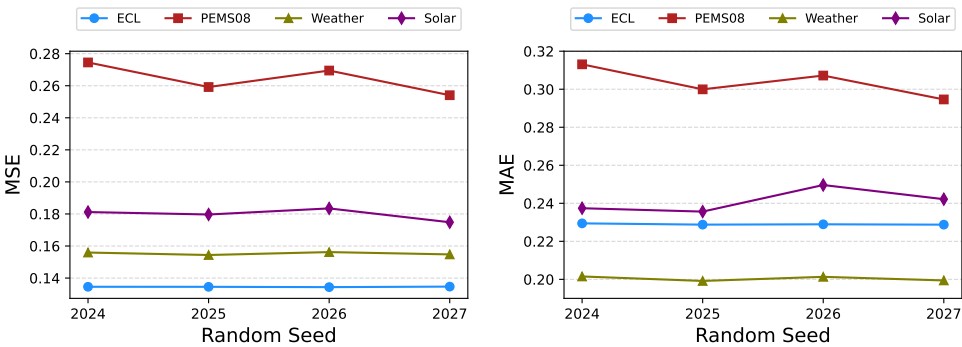

Figure 5: Performance of GTR under different random seeds on several datasets. The look-back length $T$ and forecasting horizon $S$ are fixed at 96.

Table 10: Full results of different models with the look-back length $T$=96. The reported results with standard deviation of GTR are averaged from 5 runs (with different random seeds of {2025, 2026, 2027, 2028, 2029}). The best results are highlighted in **bold**, while the second-best results are underlined.

| Model | | TimeMixer (2024c) | | iTransformer (2024b) | | CycleNet (2024) | | TimeXer (2024d) | | GTR (Ours) | |
|---|---|---|---|---|---|---|---|---|---|---|---|
| Metric | | MSE | MAE | MSE | MAE | MSE | MAE | MSE | MAE | MSE | MAE |
| ETTh1 | 96 | 0.375 | 0.400 | 0.386 | 0.405 | 0.375 | 0.395 | 0.382 | 0.403 | **0.367** ± 0.002 | **0.391** ± 0.001 |
| | 192 | 0.429 | **0.421** | 0.441 | 0.436 | 0.436 | 0.428 | 0.429 | 0.435 | **0.420** ± 0.002 | 0.421 ± 0.001 |
| | 336 | 0.484 | 0.458 | 0.487 | 0.458 | 0.496 | 0.455 | **0.468** | 0.448 | 0.476 ± 0.006 | **0.447** ± 0.001 |
| | 720 | 0.498 | 0.482 | 0.503 | 0.491 | 0.520 | 0.484 | **0.469** | **0.461** | 0.493 ± 0.021 | 0.476 ± 0.011 |
| | Avg | 0.447 | 0.440 | 0.454 | 0.448 | 0.457 | 0.441 | **0.437** | 0.437 | 0.439 ± 0.007 | **0.434** ± 0.003 |
| ETTh2 | 96 | 0.289 | 0.341 | 0.297 | 0.349 | 0.298 | 0.344 | **0.286** | **0.338** | 0.290 ± 0.002 | 0.342 ± 0.001 |
| | 192 | 0.372 | 0.392 | 0.380 | 0.400 | 0.372 | 0.396 | 0.363 | **0.389** | **0.362** ± 0.002 | 0.389 ± 0.001 |
| | 336 | **0.386** | **0.414** | 0.428 | 0.432 | 0.431 | 0.439 | 0.414 | 0.423 | 0.414 ± 0.003 | 0.429 ± 0.002 |
| | 720 | 0.412 | 0.434 | 0.427 | 0.445 | 0.450 | 0.458 | **0.408** | **0.432** | 0.423 ± 0.008 | 0.442 ± 0.004 |
| | Avg | **0.365** | **0.395** | 0.383 | 0.407 | 0.388 | 0.409 | 0.368 | 0.396 | 0.372 ± 0.003 | 0.400 ± 0.002 |
| ETTm1 | 96 | 0.320 | 0.357 | 0.334 | 0.368 | 0.319 | 0.360 | 0.318 | 0.356 | **0.305** ± 0.001 | **0.349** ± 0.001 |
| | 192 | 0.361 | 0.381 | 0.377 | 0.391 | 0.360 | 0.381 | 0.362 | 0.383 | **0.349** ± 0.001 | **0.375** ± 0.001 |
| | 336 | 0.390 | 0.404 | 0.426 | 0.420 | 0.389 | 0.403 | 0.395 | 0.407 | **0.380** ± 0.001 | **0.398** ± 0.001 |
| | 720 | 0.454 | 0.441 | 0.491 | 0.459 | 0.447 | 0.441 | 0.452 | 0.441 | **0.435** ± 0.001 | **0.436** ± 0.001 |
| | Avg | 0.381 | 0.396 | 0.407 | 0.410 | 0.379 | 0.396 | 0.382 | 0.397 | **0.367** ± 0.001 | **0.389** ± 0.001 |
| ETTm2 | 96 | 0.175 | 0.258 | 0.180 | 0.264 | **0.163** | **0.246** | 0.171 | 0.256 | 0.168 ± 0.001 | 0.249 ± 0.001 |
| | 192 | 0.237 | 0.299 | 0.250 | 0.309 | **0.229** | **0.290** | 0.237 | 0.299 | 0.232 ± 0.001 | 0.292 ± 0.001 |
| | 336 | 0.298 | 0.340 | 0.311 | 0.348 | **0.284** | **0.327** | 0.296 | 0.338 | 0.287 ± 0.001 | 0.330 ± 0.001 |
| | 720 | 0.391 | 0.396 | 0.412 | 0.407 | 0.389 | 0.391 | 0.392 | 0.394 | **0.386** ± 0.001 | **0.390** ± 0.001 |
| | Avg | 0.275 | 0.323 | 0.288 | 0.332 | **0.266** | **0.314** | 0.274 | 0.322 | 0.268 ± 0.001 | 0.315 ± 0.001 |
| Electricity | 96 | 0.153 | 0.247 | 0.148 | 0.240 | 0.136 | **0.229** | 0.140 | 0.242 | **0.134** ± 0.001 | 0.229 ± 0.001 |
| | 192 | 0.166 | 0.256 | 0.162 | 0.253 | **0.152** | **0.244** | 0.157 | 0.256 | 0.152 ± 0.001 | 0.245 ± 0.001 |
| | 336 | 0.185 | 0.277 | 0.178 | 0.269 | **0.170** | **0.264** | 0.176 | 0.275 | 0.171 ± 0.001 | 0.264 ± 0.001 |
| | 720 | 0.225 | 0.310 | 0.225 | 0.317 | 0.212 | **0.299** | 0.211 | 0.306 | **0.208** ± 0.001 | 0.300 ± 0.001 |
| | Avg | 0.182 | 0.273 | 0.178 | 0.270 | 0.168 | **0.259** | 0.171 | 0.270 | **0.166** ± 0.001 | 0.260 ± 0.001 |
| Solar-Energy | 96 | 0.189 | 0.259 | 0.203 | 0.237 | 0.190 | 0.247 | 0.215 | 0.295 | **0.176** ± 0.004 | **0.235** ± 0.008 |
| | 192 | 0.222 | 0.283 | 0.233 | 0.261 | 0.210 | 0.266 | 0.236 | 0.301 | **0.193** ± 0.004 | **0.244** ± 0.007 |
| | 336 | 0.231 | 0.292 | 0.248 | 0.273 | 0.217 | 0.266 | 0.252 | 0.307 | **0.201** ± 0.004 | **0.250** ± 0.008 |
| | 720 | 0.223 | 0.285 | 0.249 | 0.275 | 0.223 | 0.266 | 0.244 | 0.305 | **0.205** ± 0.004 | **0.251** ± 0.003 |
| | Avg | 0.216 | 0.280 | 0.233 | 0.262 | 0.210 | 0.261 | 0.237 | 0.302 | **0.194** ± 0.004 | **0.245** ± 0.006 |
| Weather | 96 | 0.163 | 0.209 | 0.174 | 0.214 | 0.158 | 0.203 | 0.157 | 0.205 | **0.154** ± 0.001 | **0.200** ± 0.001 |
| | 192 | 0.208 | 0.250 | 0.221 | 0.254 | 0.207 | 0.247 | 0.204 | 0.247 | **0.202** ± 0.001 | **0.244** ± 0.001 |
| | 336 | **0.251** | **0.287** | 0.278 | 0.296 | 0.262 | 0.289 | 0.261 | 0.290 | 0.259 ± 0.001 | 0.287 ± 0.001 |
| | 720 | **0.339** | **0.341** | 0.358 | 0.349 | 0.344 | 0.344 | 0.340 | 0.341 | 0.341 ± 0.001 | 0.342 ± 0.001 |
| | Avg | 0.240 | 0.272 | 0.258 | 0.278 | 0.243 | 0.271 | 0.241 | 0.271 | **0.239** ± 0.001 | **0.268** ± 0.001 |

## C.4 FULL RESULTS WITH LONGER LOOK-BACK WINDOW

With the recent development of model light weighting techniques, particularly the adoption of channel-independent strategies (first applied in DLinear (Zeng et al., 2023) and PatchTST (Nie et al., 2023)), more models have started to experiment with longer look-back windows in pursuit of higher predictive accuracy. For instance, DLinear and PatchTST default to using look-back windows of $T = 336$, while RAFT (Han et al., 2025) and SparseTSF (Lin et al., 2025b) default to using $T = 720$. To explore GTR's performance with longer look-back windows, we compared GTR with these advanced models using their respective default, longer look-back windows in Table 11.

It can be seen that GTR maintains a significant performance advantage even when compared against state-of-the-art baselines utilizing extended historical contexts. Specifically, GTR achieves the best MSE performance in 19 out of 32 metrics under the $T = 336$ setting and 15 out of 32 metrics under the $T = 720$ setting, consistently surpassing strong competitors like PatchTST, SparseTSF, and

Table 11: Full results of different models with longer look-back lengths $T \in \{336, 720\}$. The best results are highlighted in **bold** and the second best are underlined.

| Lookback | | T = 336 | | | | | | | | T = 720 | | | | | | | |
|---|---|---|---|---|---|---|---|---|---|---|---|---|---|---|---|---|---|
| Model | | DLinear (2023) | | PatchTST (2023) | | CycleNet (2024) | | GTR (Ours) | | RAFT (2025) | | SparseTSF (2025b) | | CycleNet (2024) | | GTR (Ours) | |
| Metric | | MSE | MAE | MSE | MAE | MSE | MAE | MSE | MAE | MSE | MAE | MSE | MAE | MSE | MAE | MSE | MAE |
| ETTh1 | 96 | 0.374 | 0.398 | 0.385 | 0.405 | 0.374 | 0.396 | 0.369 | 0.395 | 0.367 | 0.397 | 0.362 | 0.388 | 0.379 | 0.403 | 0.368 | 0.400 |
| | 192 | 0.430 | 0.440 | 0.414 | 0.421 | 0.406 | 0.415 | 0.409 | 0.420 | 0.411 | 0.427 | 0.403 | 0.411 | 0.416 | 0.425 | 0.409 | 0.424 |
| | 336 | 0.442 | 0.445 | 0.440 | 0.440 | 0.431 | 0.430 | 0.432 | 0.434 | 0.436 | 0.442 | 0.434 | 0.428 | 0.447 | 0.445 | 0.443 | 0.449 |
| | 720 | 0.497 | 0.507 | 0.456 | 0.470 | 0.450 | 0.464 | 0.451 | 0.463 | 0.467 | 0.478 | 0.426 | 0.447 | 0.477 | 0.483 | 0.464 | 0.476 |
| | Avg | 0.436 | 0.448 | 0.424 | 0.434 | 0.415 | 0.426 | 0.415 | 0.428 | 0.420 | 0.436 | 0.406 | 0.419 | 0.430 | 0.439 | 0.421 | 0.437 |
| ETTh2 | 96 | 0.281 | 0.347 | 0.275 | 0.337 | 0.279 | 0.341 | 0.274 | 0.341 | 0.276 | 0.344 | 0.294 | 0.346 | 0.271 | 0.337 | 0.275 | 0.343 |
| | 192 | 0.367 | 0.404 | 0.338 | 0.379 | 0.342 | 0.385 | 0.340 | 0.385 | 0.347 | 0.393 | 0.339 | 0.377 | 0.332 | 0.380 | 0.340 | 0.388 |
| | 336 | 0.438 | 0.454 | 0.365 | 0.398 | 0.371 | 0.413 | 0.363 | 0.409 | 0.376 | 0.425 | 0.359 | 0.397 | 0.362 | 0.408 | 0.377 | 0.416 |
| | 720 | 0.598 | 0.549 | 0.391 | 0.429 | 0.426 | 0.451 | 0.405 | 0.439 | 0.436 | 0.473 | 0.383 | 0.424 | 0.415 | 0.449 | 0.406 | 0.442 |
| | Avg | 0.421 | 0.439 | 0.342 | 0.386 | 0.355 | 0.398 | 0.345 | 0.393 | 0.359 | 0.409 | 0.344 | 0.386 | 0.345 | 0.394 | 0.349 | 0.397 |
| ETTm1 | 96 | 0.307 | 0.350 | 0.291 | 0.343 | 0.299 | 0.348 | 0.283 | 0.337 | 0.302 | 0.349 | 0.312 | 0.354 | 0.307 | 0.353 | 0.290 | 0.345 |
| | 192 | 0.340 | 0.373 | 0.334 | 0.370 | 0.334 | 0.367 | 0.323 | 0.365 | 0.329 | 0.367 | 0.347 | 0.376 | 0.337 | 0.371 | 0.328 | 0.371 |
| | 336 | 0.377 | 0.397 | 0.367 | 0.392 | 0.368 | 0.386 | 0.355 | 0.384 | 0.355 | 0.383 | 0.367 | 0.386 | 0.364 | 0.387 | 0.357 | 0.388 |
| | 720 | 0.433 | 0.433 | 0.422 | 0.426 | 0.417 | 0.414 | 0.420 | 0.422 | 0.406 | 0.413 | 0.419 | 0.413 | 0.410 | 0.411 | 0.417 | 0.420 |
| | Avg | 0.364 | 0.388 | 0.354 | 0.383 | 0.355 | 0.379 | 0.345 | 0.377 | 0.348 | 0.378 | 0.361 | 0.382 | 0.355 | 0.381 | 0.348 | 0.381 |
| ETTm2 | 96 | 0.165 | 0.257 | 0.164 | 0.254 | 0.159 | 0.247 | 0.161 | 0.252 | 0.164 | 0.256 | 0.163 | 0.252 | 0.159 | 0.249 | 0.169 | 0.258 |
| | 192 | 0.227 | 0.307 | 0.221 | 0.293 | 0.214 | 0.286 | 0.221 | 0.292 | 0.219 | 0.296 | 0.217 | 0.290 | 0.214 | 0.289 | 0.220 | 0.295 |
| | 336 | 0.304 | 0.362 | 0.276 | 0.328 | 0.269 | 0.322 | 0.278 | 0.326 | 0.275 | 0.336 | 0.270 | 0.327 | 0.268 | 0.326 | 0.270 | 0.328 |
| | 720 | 0.431 | 0.441 | 0.366 | 0.383 | 0.363 | 0.382 | 0.359 | 0.388 | 0.359 | 0.392 | 0.352 | 0.379 | 0.353 | 0.384 | 0.355 | 0.389 |
| | Avg | 0.282 | 0.342 | 0.257 | 0.315 | 0.251 | 0.309 | 0.254 | 0.314 | 0.254 | 0.320 | 0.251 | 0.312 | 0.249 | 0.312 | 0.253 | 0.317 |
| Electricity | 96 | 0.140 | 0.237 | 0.131 | 0.225 | 0.128 | 0.223 | 0.127 | 0.222 | 0.133 | 0.232 | 0.138 | 0.233 | 0.128 | 0.223 | 0.128 | 0.226 |
| | 192 | 0.153 | 0.250 | 0.148 | 0.240 | 0.144 | 0.237 | 0.147 | 0.242 | 0.149 | 0.247 | 0.151 | 0.244 | 0.143 | 0.237 | 0.146 | 0.244 |
| | 336 | 0.169 | 0.267 | 0.165 | 0.259 | 0.160 | 0.254 | 0.165 | 0.260 | 0.161 | 0.259 | 0.166 | 0.260 | 0.159 | 0.254 | 0.162 | 0.261 |
| | 720 | 0.203 | 0.299 | 0.202 | 0.291 | 0.198 | 0.287 | 0.199 | 0.292 | 0.197 | 0.297 | 0.205 | 0.293 | 0.197 | 0.287 | 0.197 | 0.295 |
| | Avg | 0.166 | 0.263 | 0.162 | 0.254 | 0.158 | 0.250 | 0.159 | 0.254 | 0.160 | 0.259 | 0.165 | 0.258 | 0.157 | 0.250 | 0.158 | 0.256 |
| Solar-Energy | 96 | 0.222 | 0.292 | 0.190 | 0.278 | 0.200 | 0.250 | 0.180 | 0.237 | 0.192 | 0.251 | 0.195 | 0.243 | 0.194 | 0.255 | 0.178 | 0.242 |
| | 192 | 0.249 | 0.313 | 0.206 | 0.252 | 0.221 | 0.261 | 0.196 | 0.254 | 0.247 | 0.323 | 0.215 | 0.251 | 0.205 | 0.251 | 0.198 | 0.260 |
| | 336 | 0.268 | 0.327 | 0.217 | 0.254 | 0.236 | 0.272 | 0.201 | 0.257 | 0.240 | 0.300 | 0.232 | 0.262 | 0.218 | 0.257 | 0.202 | 0.264 |
| | 720 | 0.271 | 0.326 | 0.219 | 0.255 | 0.245 | 0.277 | 0.205 | 0.258 | 0.246 | 0.311 | 0.237 | 0.263 | 0.239 | 0.278 | 0.209 | 0.269 |
| | Avg | 0.253 | 0.315 | 0.208 | 0.260 | 0.226 | 0.265 | 0.196 | 0.252 | 0.231 | 0.296 | 0.220 | 0.256 | 0.214 | 0.260 | 0.197 | 0.259 |
| Weather | 96 | 0.174 | 0.235 | 0.155 | 0.204 | 0.167 | 0.221 | 0.147 | 0.199 | 0.165 | 0.222 | 0.169 | 0.223 | 0.164 | 0.220 | 0.146 | 0.200 |
| | 192 | 0.219 | 0.281 | 0.195 | 0.242 | 0.212 | 0.258 | 0.192 | 0.240 | 0.211 | 0.264 | 0.214 | 0.262 | 0.209 | 0.258 | 0.192 | 0.244 |
| | 336 | 0.264 | 0.317 | 0.249 | 0.283 | 0.260 | 0.293 | 0.243 | 0.281 | 0.260 | 0.302 | 0.257 | 0.293 | 0.255 | 0.292 | 0.243 | 0.283 |
| | 720 | 0.324 | 0.363 | 0.321 | 0.334 | 0.328 | 0.339 | 0.324 | 0.337 | 0.327 | 0.355 | 0.321 | 0.340 | 0.320 | 0.338 | 0.317 | 0.336 |
| | Avg | 0.245 | 0.299 | 0.230 | 0.266 | 0.242 | 0.278 | 0.227 | 0.264 | 0.241 | 0.286 | 0.240 | 0.280 | 0.237 | 0.277 | 0.225 | 0.266 |
| $1^{st}$ Count | | 0 | 0 | 4 | 9 | 13 | 13 | 19 | 13 | 5 | 3 | 9 | 13 | 11 | 13 | 15 | 7 |

RAFT. Notably, on the Solar-Energy dataset with $T = 336$, GTR reduces MSE by 13.2% compared to the CycleNet, and on Weather with $T = 720$, it outperforms RAFT by 6.7% in MSE.

Crucially, although the architectural design of GTR primarily targets scenarios with restricted historical contexts (Figure 3), it exhibits remarkable adaptability when scaled to extended horizons. As evidenced by the results, GTR does not merely function as a remedial solution for short inputs but continues to deliver competitive, state-of-the-art performance under long look-back settings. This confirms that GTR's mechanism for explicit global cycle retrieval is robust and effective.

## C.5 COMPARISON WITH TIMESTAMP-BASED APPROACHES

Distinct from methods that learn periodicity directly from historical data, another prevalent paradigm in long-term forecasting leverages explicit timestamp information (e.g., time-of-day, day-of-week) to model global temporal dependencies (Wang et al., 2024a; Yang et al., 2025a). These approaches typically operate as modular plugins, designed to augment existing backbones by injecting global guidance derived from calendar features. Specifically, they model timestamps to capture global contexts and adaptively fuse these signals with local observations.

To comprehensively evaluate the effectiveness of our proposed GTR module, we conducted a comparative study using GLAFF (Wang et al., 2024a) as a representative baseline. We integrated both our proposed GTR module and the GLAFF module into two distinct backbones, DLinear (Zeng et al., 2023) and iTransformer (Liu et al., 2024b), and evaluated their performance on the Electricity and Weather datasets. The detailed results are presented in Table 12.

Table 12: Ablation study of GTR and GLAFF modules with the look-back length $T = 96$. *All results in this table are reproduced using the official GLAFF implementation*. We compare the performance (MSE, MAE) and efficiency (Training Time, Memory Cost) across four prediction horizons ($T \in \{96, 192, 336, 720\}$) on Electricity and Weather datasets. The best performance in each setting is highlighted in **bold**.

| Model | Horizon | Modules | | Electricity | | | | Weather | | | |
|---|---|---|---|---|---|---|---|---|---|---|---|
| | | GTR | GLAFF | MSE | MAE | Time(s) | Mem(MB) | MSE | MAE | Time(s) | Mem(MB) |
| DLinear | 96 | ✓ | ✗ | 0.2073 | 0.2948 | 52.04 | **0.31** | **0.1974** | **0.2572** | **26.08** | **0.12** |
| | | ✗ | ✓ | **0.1653** | **0.2602** | **23.85** | 4.85 | 0.2070 | 0.2653 | 38.89 | 4.36 |
| | 192 | ✓ | ✗ | 0.2042 | 0.2939 | 46.64 | **0.38** | **0.2410** | **0.2963** | **30.78** | **0.19** |
| | | ✗ | ✓ | **0.1851** | **0.2747** | **32.47** | 4.92 | 0.2530 | 0.3069 | 39.17 | 4.43 |
| | 336 | ✓ | ✗ | 0.2197 | 0.3117 | 36.72 | **0.49** | **0.2871** | **0.3347** | **15.94** | **0.30** |
| | | ✗ | ✓ | **0.2179** | **0.3053** | **22.35** | 5.03 | 0.2955 | 0.3397 | 37.35 | 4.54 |
| | 720 | ✓ | ✗ | **0.2569** | **0.3448** | **19.09** | **0.77** | **0.3554** | **0.3887** | **8.68** | **0.58** |
| | | ✗ | ✓ | 0.2904 | 0.3553 | 19.13 | 5.31 | 0.4542 | 0.4180 | 27.99 | 4.82 |
| iTransformer | 96 | ✓ | ✗ | **0.1476** | **0.2411** | 86.33 | **48.72** | **0.1731** | **0.2168** | 61.77 | **48.53** |
| | | ✗ | ✓ | 0.1504 | 0.2434 | **45.34** | 53.26 | 0.2028 | 0.2410 | 78.12 | 52.77 |
| | 192 | ✓ | ✗ | **0.1680** | **0.2610** | 74.71 | **48.91** | **0.2204** | **0.2615** | 55.42 | **48.72** |
| | | ✗ | ✓ | 0.1922 | 0.2685 | **36.50** | 53.45 | 0.2476 | 0.2797 | **54.04** | 52.96 |
| | 336 | ✓ | ✗ | **0.1838** | **0.2781** | 36.83 | **49.19** | **0.2829** | **0.3061** | 29.95 | **49.00** |
| | | ✗ | ✓ | 0.2343 | 0.2968 | **34.68** | 53.73 | 0.3398 | 0.3322 | 51.46 | 53.24 |
| | 720 | ✓ | ✗ | **0.2178** | **0.3047** | **34.97** | **49.95** | **0.3711** | **0.3614** | **27.25** | **49.75** |
| | | ✗ | ✓ | 0.5529 | 0.4208 | 103.07 | 54.00 | 0.6578 | 0.4316 | 51.02 | 54.00 |

**Prediction Accuracy and Robustness.** As shown in Table 12, the GTR module demonstrates superior performance and robustness compared to the timestamp-based GLAFF, particularly in long-term forecasting scenarios. While timestamp-based methods can capture explicit cyclic patterns in shorter horizons (e.g., DLinear on Electricity at $T = 96$), they struggle to generalize to longer prediction windows due to the potential overfitting of rigid calendar features. Notably, at the prediction horizon of $T = 720$, the performance of GLAFF degrades significantly on both datasets (e.g., on iTransformer, the MSE on Electricity spikes to 0.5529). In contrast, GTR maintains stable and accurate predictions (MSE 0.2178), indicating that learning periodicity directly from time-series data—rather than relying on external timestamps—yields better robustness for long-term modeling.

**Computational Efficiency.** Beyond prediction accuracy, GTR exhibits a significant advantage in computational efficiency. Timestamp-based approaches like GLAFF typically require additional embedding layers and complex attention or mixing mechanisms to process high-dimensional timestamp features, leading to increased memory usage and training time. GTR incurs negligible memory overhead compared to the backbone models (e.g., ∼0.3 MB vs. ∼4.8 MB on DLinear). This lightweight characteristic makes GTR a more practical solution for real-world deployments where computational resources are constrained, achieving a better trade-off between performance and cost.

## C.6 VISUALIZATION RESULTS

To provide a clear and comprehensive comparison among different models, we present supplementary prediction visualizations on the PEMS datasets in the accompanying figures (6; 7; 8; 9;). Among all evaluated models, GTR consistently demonstrates the highest precision in capturing future traffic series variations and exhibits superior overall forecasting performance.

Furthermore, we conduct a direct visual comparison between GTR and its underlying MLP backbone model. The results reveal a striking improvement: GTR significantly enhances the fitting accuracy of the original MLP, effectively mitigating its tendency to oversmooth temporal dynamics and miss subtle yet critical traffic patterns. Notably, across multiple visualization scenarios, the input sequences exhibit little to no pronounced periodicity—yet GTR is still able to accurately capture and forecast the underlying future trends. This highlights GTR's robustness in modeling complex, non-stationary traffic dynamics even in the absence of strong periodic patterns, further underscoring the effectiveness of its architectural design in learning global temporal representations.

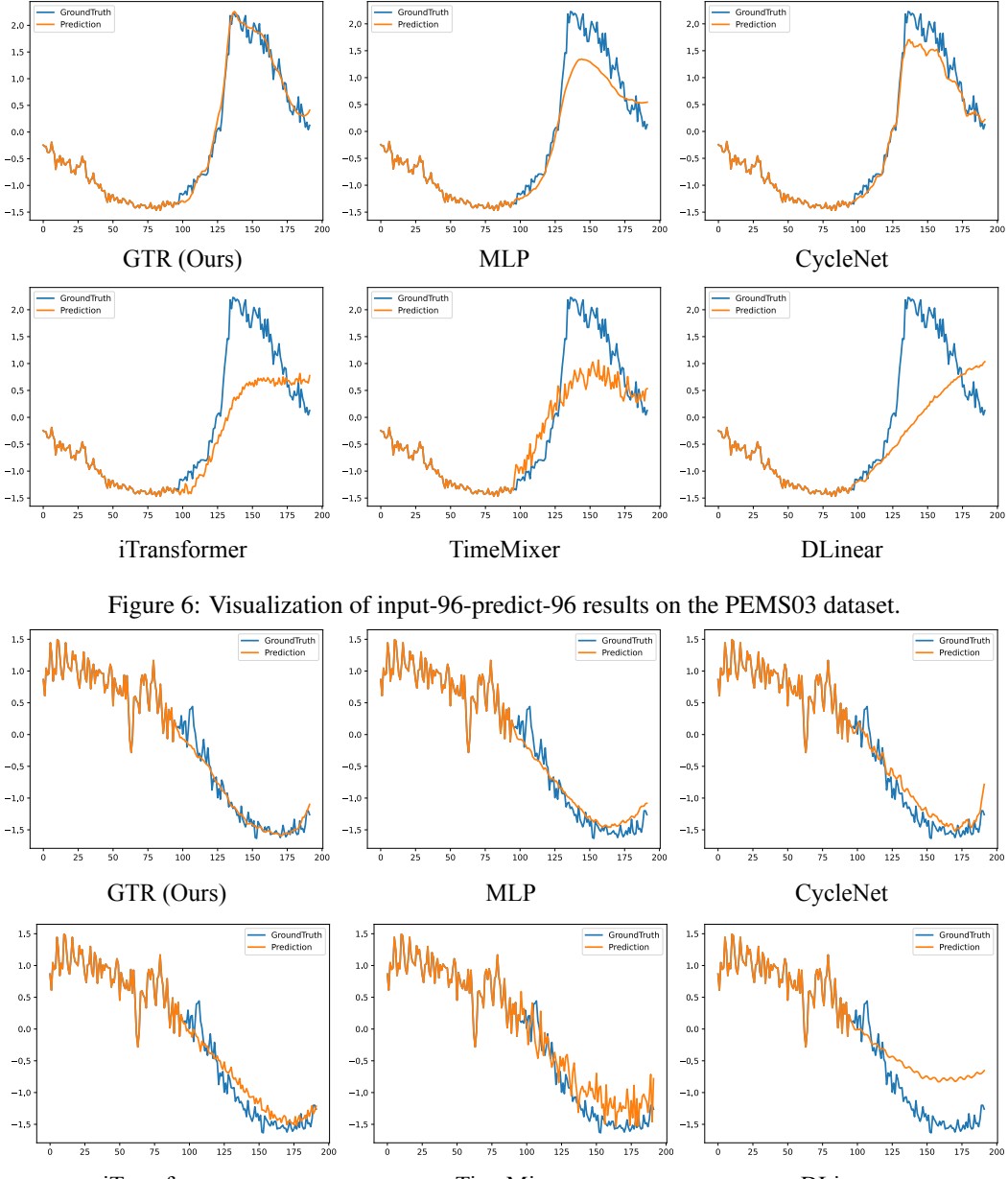

Figure 6: Visualization of input-96-predict-96 results on the PEMS03 dataset.

Figure 7: Visualization of input-96-predict-96 results on the PEMS04 dataset.

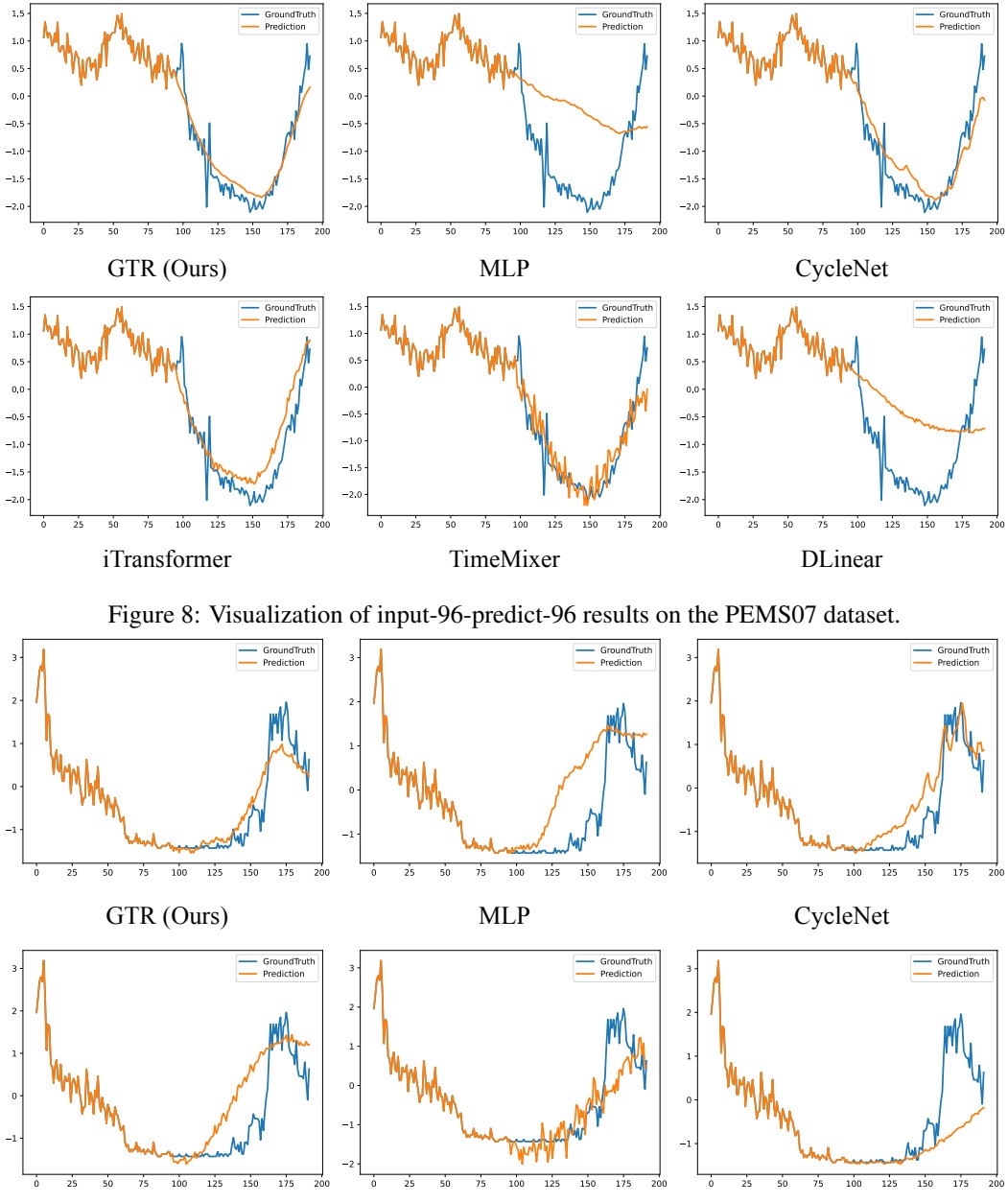

Figure 8: Visualization of input-96-predict-96 results on the PEMS07 dataset.

Figure 9: Visualization of input-96-predict-96 results on the PEMS08 dataset.

## D    THEORETICAL ANALYSIS

In this section, we provide a rigorous theoretical analysis of the GTR module. We demonstrate that GTR reduces the estimation error of multivariate correlations by leveraging global temporal information, using a Bayesian estimation framework with strict mathematical derivation.

### D.1    PROBLEM SETUP AND ASSUMPTIONS

Consider two variables $n$ and $m$ in a multivariate time series. Let $Y_n$ and $Y_m$ represent the ground-truth periodic patterns of these variables, which are zero-mean random variables with variance $\sigma_Y^2$ and correlation $\rho = \mathbb{E}[Y_n Y_m]/\sigma_Y^2$.

We make the following assumptions:

1. **Observation Error:** The observed segment $x_n = Y_n + \eta_n$, where $\eta_n \sim \mathcal{N}(0, \sigma_\eta^2)$ represents the error term caused by non-stationary phenomena.

2. **Global Embedding Error:** The global temporal embedding $Q_n = Y_n + \varepsilon_n$, where $\varepsilon_n \sim \mathcal{N}(0, \sigma_\varepsilon^2)$ is independent embedding error.

3. **Independence:** All error terms $\eta_n, \eta_m, \varepsilon_n, \varepsilon_m$ are mutually independent.

### D.2    GTR AS A BAYESIAN ESTIMATOR

During the inference stage of GTR, the operation of combining observed segments $x_n$ and global embeddings $Q_n$ is a pure linear combination:

$$z_n = \frac{\sigma_\varepsilon^2 x_n + \sigma_\eta^2 Q_n}{\sigma_\eta^2 + \sigma_\varepsilon^2} \tag{8}$$

This formulation corresponds to the posterior mean estimator in a Bayesian framework, where:

- $x_n$ represents the likelihood of observing $Y_n$
- $Q_n$ represents the prior information about $Y_n$
- The weights $\sigma_\varepsilon^2$ and $\sigma_\eta^2$ are proportional to the inverse of the noise variances

### D.3    THEORETICAL RESULT

**Theorem D.1.** *Given the above assumptions, if $\sigma_\varepsilon^2 < \sigma_\eta^2$, then:*

$$|\text{corr}(z_n, z_m) - \rho| < |\text{corr}(x_n, x_m) - \rho| \tag{9}$$

*where $\text{corr}(x_n, x_m)$ is the correlation of raw observations, and $\text{corr}(z_n, z_m)$ is the correlation after GTR module processing.*

*Proof.* First, we compute the correlation of raw observations:

$$\begin{aligned}
\mathbb{E}[x_n x_m] &= \mathbb{E}[(Y_n + \eta_n)(Y_m + \eta_m)] \\
&= \mathbb{E}[Y_n Y_m] + \mathbb{E}[Y_n \eta_m] + \mathbb{E}[\eta_n Y_m] + \mathbb{E}[\eta_n \eta_m] \\
&= \rho \sigma_Y^2 + 0 + 0 + \sigma_\eta^2 \\
&= \rho \sigma_Y^2 + \sigma_\eta^2
\end{aligned}$$

$$\begin{aligned}
\text{Var}(x_n) &= \mathbb{E}[x_n^2] - (\mathbb{E}[x_n])^2 \\
&= \mathbb{E}[(Y_n + \eta_n)^2] \\
&= \mathbb{E}[Y_n^2] + \mathbb{E}[\eta_n^2] \\
&= \sigma_Y^2 + \sigma_\eta^2
\end{aligned}$$

Therefore:

$$\text{corr}(x_n, x_m) = \frac{\mathbb{E}[x_n x_m]}{\sqrt{\text{Var}(x_n)\text{Var}(x_m)}} = \frac{\rho \sigma_Y^2 + \sigma_\eta^2}{\sigma_Y^2 + \sigma_\eta^2} \tag{10}$$

The estimation error for raw observations is:

$$|\text{corr}(x_n, x_m) - \rho| = \left| \frac{\rho \sigma_Y^2 + \sigma_\eta^2}{\sigma_Y^2 + \sigma_\eta^2} - \rho \right|$$

$$= \left| \frac{\rho \sigma_Y^2 + \sigma_\eta^2 - \rho \sigma_Y^2 - \rho \sigma_\eta^2}{\sigma_Y^2 + \sigma_\eta^2} \right|$$

$$= \left| \frac{\sigma_\eta^2 (1 - \rho)}{\sigma_Y^2 + \sigma_\eta^2} \right|$$

Next, we compute the correlation after GTR processing. First, the expectation:

$$\mathbb{E}[z_n z_m] = \mathbb{E}\left[ \frac{(\sigma_\varepsilon^2 x_n + \sigma_\eta^2 Q_n)(\sigma_\varepsilon^2 x_m + \sigma_\eta^2 Q_m)}{(\sigma_\eta^2 + \sigma_\varepsilon^2)^2} \right]$$

$$= \frac{1}{(\sigma_\eta^2 + \sigma_\varepsilon^2)^2} \left[ \sigma_\varepsilon^4 \mathbb{E}[x_n x_m] + \sigma_\varepsilon^2 \sigma_\eta^2 \mathbb{E}[x_n Q_m] + \sigma_\eta^2 \sigma_\varepsilon^2 \mathbb{E}[Q_n x_m] + \sigma_\eta^4 \mathbb{E}[Q_n Q_m] \right]$$

Calculate each term:

$$\mathbb{E}[x_n x_m] = \rho \sigma_Y^2 + \sigma_\eta^2$$

$$\mathbb{E}[x_n Q_m] = \mathbb{E}[(Y_n + \eta_n)(Y_m + \varepsilon_m)] = \rho \sigma_Y^2$$

$$\mathbb{E}[Q_n x_m] = \mathbb{E}[(Y_n + \varepsilon_n)(Y_m + \eta_m)] = \rho \sigma_Y^2$$

$$\mathbb{E}[Q_n Q_m] = \mathbb{E}[(Y_n + \varepsilon_n)(Y_m + \varepsilon_m)] = \rho \sigma_Y^2 + \sigma_\varepsilon^2$$

Substituting these values:

$$\mathbb{E}[z_n z_m] = \frac{1}{(\sigma_\eta^2 + \sigma_\varepsilon^2)^2} \left[ \sigma_\varepsilon^4 (\rho \sigma_Y^2 + \sigma_\eta^2) + \sigma_\varepsilon^2 \sigma_\eta^2 (\rho \sigma_Y^2) + \sigma_\eta^2 \sigma_\varepsilon^2 (\rho \sigma_Y^2) + \sigma_\eta^4 (\rho \sigma_Y^2 + \sigma_\varepsilon^2) \right]$$

$$= \frac{1}{(\sigma_\eta^2 + \sigma_\varepsilon^2)^2} \left[ \rho \sigma_Y^2 (\sigma_\varepsilon^4 + 2\sigma_\varepsilon^2 \sigma_\eta^2 + \sigma_\eta^4) + \sigma_\varepsilon^4 \sigma_\eta^2 + \sigma_\eta^4 \sigma_\varepsilon^2 \right]$$

$$= \frac{1}{(\sigma_\eta^2 + \sigma_\varepsilon^2)^2} \left[ \rho \sigma_Y^2 (\sigma_\eta^2 + \sigma_\varepsilon^2)^2 + \sigma_\varepsilon^2 \sigma_\eta^2 (\sigma_\eta^2 + \sigma_\varepsilon^2) \right]$$

$$= \rho \sigma_Y^2 + \frac{\sigma_\varepsilon^2 \sigma_\eta^2}{\sigma_\eta^2 + \sigma_\varepsilon^2}$$

Now calculate the variance:

$$\text{Var}(z_n) = \mathbb{E}[z_n^2] - (\mathbb{E}[z_n])^2$$

$$= \mathbb{E}\left[ \left( \frac{\sigma_\varepsilon^2 x_n + \sigma_\eta^2 Q_n}{\sigma_\eta^2 + \sigma_\varepsilon^2} \right)^2 \right]$$

$$= \frac{1}{(\sigma_\eta^2 + \sigma_\varepsilon^2)^2} \left[ \sigma_\varepsilon^4 \mathbb{E}[x_n^2] + 2\sigma_\varepsilon^2 \sigma_\eta^2 \mathbb{E}[x_n Q_n] + \sigma_\eta^4 \mathbb{E}[Q_n^2] \right]$$

Calculate each term:

$$\mathbb{E}[x_n^2] = \sigma_Y^2 + \sigma_\eta^2$$

$$\mathbb{E}[Q_n^2] = \sigma_Y^2 + \sigma_\varepsilon^2$$

$$\mathbb{E}[x_n Q_n] = \mathbb{E}[(Y_n + \eta_n)(Y_n + \varepsilon_n)] = \sigma_Y^2$$

Substituting these values:

$$\text{Var}(z_n) = \frac{1}{(\sigma_\eta^2 + \sigma_\varepsilon^2)^2}\left[\sigma_\varepsilon^4(\sigma_Y^2 + \sigma_\eta^2) + 2\sigma_\varepsilon^2\sigma_\eta^2\sigma_Y^2 + \sigma_\eta^4(\sigma_Y^2 + \sigma_\varepsilon^2)\right]$$

$$= \frac{1}{(\sigma_\eta^2 + \sigma_\varepsilon^2)^2}\left[\sigma_Y^2(\sigma_\varepsilon^4 + 2\sigma_\varepsilon^2\sigma_\eta^2 + \sigma_\eta^4) + \sigma_\varepsilon^4\sigma_\eta^2 + \sigma_\eta^4\sigma_\varepsilon^2\right]$$

$$= \frac{1}{(\sigma_\eta^2 + \sigma_\varepsilon^2)^2}\left[\sigma_Y^2(\sigma_\eta^2 + \sigma_\varepsilon^2)^2 + \sigma_\varepsilon^2\sigma_\eta^2(\sigma_\eta^2 + \sigma_\varepsilon^2)\right]$$

$$= \sigma_Y^2 + \frac{\sigma_\varepsilon^2\sigma_\eta^2}{\sigma_\eta^2 + \sigma_\varepsilon^2}$$

Therefore, the correlation after GTR processing is:

$$\text{corr}(z_n, z_m) = \frac{\mathbb{E}[z_n z_m]}{\sqrt{\text{Var}(z_n)\text{Var}(z_m)}}$$

$$= \frac{\rho\sigma_Y^2 + \frac{\sigma_\varepsilon^2\sigma_\eta^2}{\sigma_\eta^2 + \sigma_\varepsilon^2}}{\sigma_Y^2 + \frac{\sigma_\varepsilon^2\sigma_\eta^2}{\sigma_\eta^2 + \sigma_\varepsilon^2}}$$

The estimation error for GTR-processed data is:

$$|\text{corr}(z_n, z_m) - \rho| = \left|\frac{\rho\sigma_Y^2 + \frac{\sigma_\varepsilon^2\sigma_\eta^2}{\sigma_\eta^2 + \sigma_\varepsilon^2}}{\sigma_Y^2 + \frac{\sigma_\varepsilon^2\sigma_\eta^2}{\sigma_\eta^2 + \sigma_\varepsilon^2}} - \rho\right|$$

$$= \left|\frac{\rho\sigma_Y^2(\sigma_\eta^2 + \sigma_\varepsilon^2) + \sigma_\varepsilon^2\sigma_\eta^2 - \rho\sigma_Y^2(\sigma_\eta^2 + \sigma_\varepsilon^2) - \rho\sigma_\varepsilon^2\sigma_\eta^2}{\sigma_Y^2(\sigma_\eta^2 + \sigma_\varepsilon^2) + \sigma_\varepsilon^2\sigma_\eta^2}\right|$$

$$= \left|\frac{\sigma_\varepsilon^2\sigma_\eta^2(1 - \rho)}{\sigma_Y^2(\sigma_\eta^2 + \sigma_\varepsilon^2) + \sigma_\varepsilon^2\sigma_\eta^2}\right|$$

Now, we compare the two errors:

$$\frac{|\text{corr}(z_n, z_m) - \rho|}{|\text{corr}(x_n, x_m) - \rho|} = \frac{\frac{\sigma_\varepsilon^2\sigma_\eta^2|1-\rho|}{\sigma_Y^2(\sigma_\eta^2 + \sigma_\varepsilon^2) + \sigma_\varepsilon^2\sigma_\eta^2}}{\frac{\sigma_\eta^2|1-\rho|}{\sigma_Y^2 + \sigma_\eta^2}}$$

$$= \frac{\sigma_\varepsilon^2(\sigma_Y^2 + \sigma_\eta^2)}{\sigma_Y^2(\sigma_\eta^2 + \sigma_\varepsilon^2) + \sigma_\varepsilon^2\sigma_\eta^2}$$

We need to show this ratio is less than 1 when $\sigma_\varepsilon^2 < \sigma_\eta^2$:

$$\sigma_\varepsilon^2(\sigma_Y^2 + \sigma_\eta^2) < \sigma_Y^2(\sigma_\eta^2 + \sigma_\varepsilon^2) + \sigma_\varepsilon^2\sigma_\eta^2$$

$$\sigma_\varepsilon^2\sigma_Y^2 + \sigma_\varepsilon^2\sigma_\eta^2 < \sigma_Y^2\sigma_\eta^2 + \sigma_Y^2\sigma_\varepsilon^2 + \sigma_\varepsilon^2\sigma_\eta^2$$

$$0 < \sigma_Y^2\sigma_\eta^2$$

Since $\sigma_Y^2 > 0$ and $\sigma_\eta^2 > 0$, the inequality holds. Therefore:

$$|\text{corr}(z_n, z_m) - \rho| < |\text{corr}(x_n, x_m) - \rho| \tag{11}$$

This completes the proof. □

GTR fuses global information sources to produce more accurate correlation estimates. As shown in Figure 4, GTR systematically reduces the error in correlation estimation between variables, confirming GTR's ability to better capture global temporal dependencies.

# E   EXTENDED RELATED WORK

**MTSF Plugins.** Recent advancements in MTSF have shifted towards "Plug-and-Play" paradigms, which decompose complex modeling tasks into modular components that can be seamlessly integrated into various backbone architectures.

A primary focus of these modules is enhancing multi-correlation and inter-variable dependencies modeling. For example, to address computational efficiency in high-dimensional data, SOFTS (Han et al., 2024) incorporates the STAR module, which aggregates global information into a core representation and redistributes it to individual series to model channel interactions with linear complexity. CrossLinear (Zhou et al., 2025) was proposed as a lightweight embedding module designed to capture time-invariant dependencies between target and exogenous variables within linear predictors. TQNet (Lin et al., 2025a) utilizes TQ techniques to learns more robust multivariate correlations.

The effective utilization of timestamp information is critical for capturing the inherent seasonality and cyclic dynamics underlying time series data. While early approaches predominantly relied on handcrafted features to encode calendar timestamps, GLAFF (Wang et al., 2024a) advances learnable representations by adaptively balancing global and local temporal information to facilitate seamless integration with arbitrary forecasting backbones. Furthermore, VH-NBEATS (Yang et al., 2025a) introduces a hierarchical timestamp basis block that explicitly leverages multi-scale calendar information to capture complex hierarchical temporal effects.

Accurately capturing periodic patterns and cyclic dynamics is fundamental for long-term forecasting, yet standard models often struggle to disentangle these from complex trends without specialized mechanisms. To address this, CycleNet (Lin et al., 2024) introduces RCF technique that explicitly models inherent periodicities using learnable recurrent cycles, thereby allowing simple backbones to focus efficiently on predicting residual dynamics. STiD (Shao et al., 2022) identifies spatial–temporal indistinguishability as a key challenge in MTSF and introduces a simple MLP-based model augmented with spatial and temporal identity embeddings to achieve strong performance. Addressing the limitations of fixed decomposition kernels, Leddam (Yu et al., 2024) proposes a learnable decomposition strategy coupled with a dual attention module to adaptively separate complex intra-series variations from inter-series dependencies. Furthermore, SparseTSF (Lin et al., 2025b) utilizes a Cross-Period Sparse Forecasting technique that decouples trend and periodicity through downsampling, functioning as a structural regularization mechanism to reduce model complexity while robustly capturing cross-period trends.

Despite these methodological strides, a critical gap remains in the current landscape of plug-and-play modules. Existing approaches lack a dedicated mechanism to address the fundamental information bottleneck caused by restricted historical contexts. Specifically, when the *look-back window is shorter than the inherent cycle length*, standard models—even when augmented with current plugins—struggle to capture global periodic dynamics due to the absence of complete cycle observations. Our proposed GTR provides an efficient and universal solution that empowers host models to dynamically retrieve and leverage global temporal patterns beyond their immediate receptive field.

# F   USES OF LLMs

This work was completed without the use of any LLMs or AI-assisted writing tools.

