# OpenReview forum: "Enhancing Multivariate Time Series Forecasting with Global Temporal Retrieval"
_ICLR.cc/2026/Conference — ICLR 2026 Poster_

### Official Review · Reviewer_HtL7 · 2025-10-27

**Soundness:** 3
**Presentation:** 3
**Contribution:** 3
**Rating:** 6
**Confidence:** 4

**Summary:**

This paper proposes a method that maintains a global temporal embedding and retrieves the corresponding temporal segment based on the absolute position of the input within the global cycle.

**Strengths:**

1. The core idea is sound and empirically effective. I particularly appreciate Figure 4, which provides a clear visualization that proves the practicality of the proposed approach.

2. The reported quantitative results are strong.

**Weaknesses:**

1. **Motivation with examples:**
The main motivation - namely, that certain global patterns cannot be captured within a limited lookback window - is reasonable but remains abstract. Providing a concrete conceptual example using an existing dataset would make this motivation more convincing and accessible.

2. **Global information guarantee of the global parameter matrix $Q$:**
The method introduces a learnable global parameter matrix $Q$, intended to encode global temporal patterns. However, it is not clear how or why this matrix is guaranteed to capture such global information. While Figure 4 provides empirical evidence, a more methodological justification or theoretical explanation would strengthen the paper.

3. **Weak ablation study:**
The ablation analysis is limited. It would be informative to include:

* Variants of the temporal pattern extraction module (e.g., replacing the 2D convolution).

* Experiments without instance normalization, to isolate its contribution.

4. **Limited comparison with related retrieval-based methods:**
The paper briefly mentions retrieval-based time-series forecasting approaches but does not include direct empirical comparisons. Including at least one retrieval-based baseline would help clarify the advantage of the proposed method.

**Questions:**

1. Provide why training this global parameter matrix $Q$ works, without any design consideration (W2).
2. Enhance experiments with more ablation study and retrieval-based baselines (W3, W4).
3. The conceptual example of scenario would strengthen the motivation (W1).

---

> ### Author Response · Authors · 2025-11-23
> **Response to Reviewer HtL7**
>
> Dear Reviewer **HtL7**:
>
> We appreciate you taking the time to review our paper and provide valuable feedback. We think your comments are very valuable for our paper's improvement and have revised the paper in response to the questions you raised. Please find responses to your questions below:
>
> ---
>
> 1. **Motivation with examples:**
>     * In **Fig. 1 (Introduction)**, we present a conceptual yet real-data–based illustration using the **Electricity** dataset. The figure visualizes **288 data points** corresponding to **two week**, segmented into **24-point daily slices**.
>     * The purpose of this figure is to illustrate a key phenomenon: **global periodic patterns (e.g., weekly cycles) are clearly visible at the full-sequence scale, yet they become much less apparent within local daily windows**.
>     * Current models, however, rely solely on such local windows and therefore fail to explicitly capture these clearly existing global periodic structures, leading to suboptimal performance.
> 2. **Global information guarantee of the global parameter matrix $Q$:**
>     * We provide a formal justification in **section 3.5** and **Appendix F**: using a Bayesian posterior–mean formulation (Appendix F, **Theorem F.1**), we show that when the global embedding error ($\sigma_\epsilon^2$) is smaller than the observation error ($\sigma_\eta^2$), GTR systematically reduces correlation-estimation error.
>     * Moreover, the absolute temporal indexing ensures that gradients from samples sharing the same cycle position aggregate onto the corresponding entries of the global parameter matrix $Q$, enabling it to effectively learn global temporal structure during training.
>     * The empirical results in **Fig. 4 and Table 3** further validate this mechanism, demonstrating that $Q$ indeed captures meaningful global patterns and is essential for the observed performance gains.
> 3. **Weak ablation study**:
>     * We have added a comprehensive ablation analysis in **Appendix K.2** and **Table 10**, covering both the extraction module design and the impact of normalization.
>     * We compared our 2D Convolution against three variants: **(1) Point-wise Fusion (Concat)**, **(2) Inception Module [1]**, and **(3) 1D Convolution**. The original **2D Convolution** consistently yields superior performance (best in 19/40 MSE cases). The inferiority of the "Concat" variant highlights the necessity of the convolutional receptive field, while the 2D structure proves more robust than treating features as abstract channels in 1D convolution.
>     * We validated the contribution of Reversible Instance Normalization. Removing RevIN leads to a catastrophic performance drop on datasets with significant distribution shifts (e.g., MSE increases from 0.423 to 0.872 on ETTh2), confirming its necessity for handling non-stationarity.
>
>     **Condensed Ablation & Comparison Results (MSE):**
>
>     | Experiment | Setting (Dataset/Horizon) | Baseline / Variant | **GTR (Ours)** | Observation |
>     | :--- | :--- | :--- | :---: | :--- |
>     | **Extraction** | ETTh1 ($T=720$) | 0.529 (Concat) | **0.493** | Receptive field is critical. |
>     | **RevIN** | ETTh2 ($T=720$) | 0.872 (w/o RevIN) | **0.423** | RevIN is essential for stability. |
>
>     *(Full detailed results are provided in Appendix K.2 Table 10).*
>
> 4. **Limited comparison with related retrieval-based methods:**
> We thank the reviewer for this comment. We would like to clarify that we have included **RAFT** [2] (ICML 2025), a representative state-of-the-art retrieval-based forecasting model, as a primary baseline in our comprehensive evaluation.
>     * **Long-term Forecasting (Table 1):** GTR consistently outperforms RAFT. For example, on the **Electricity** dataset, GTR achieves an MSE of **0.166**, surpassing RAFT (0.175).
>     * **Short-term Forecasting (Table 2):** The advantage is even more pronounced. On the **PEMS03** dataset, GTR reduces MSE by roughly **39.6%** compared to RAFT (0.087 vs. 0.144).
>     * **Long Look-back Windows (Table 11):** Even when RAFT utilizes its **default long look-back window** ($T=720$), GTR maintains a competitive edge, achieving lower MSE on datasets like **Weather** (0.317 vs. 0.327) and **ETTh2** (0.406 vs. 0.436).
>
> These results confirm that GTR delivers strong, state-of-the-art performance compared to existing retrieval-based methods.
>
> [1] Going Deeper with Convolutions
>
> [2] Retrieval Augmented Time Series Forecasting
>
> ---
>
> Once again, thank you for reviewing the paper. We think we have solved your problem as much as possible in rebuttal, If you have any further questions, please do not hesitate to contact us.

---

> > ### Comment · Reviewer_HtL7 · 2025-11-25
> >
> > Thank you for the clear and detailed rebuttal. I appreciate the clarifications and have increased my score accordingly.

---

> > > ### Author Response · Authors · 2025-11-28
> > >
> > > We sincerely appreciate the reviewer's strong support for our work.

---

### Official Review · Reviewer_L6pR · 2025-10-30

**Soundness:** 3
**Presentation:** 3
**Contribution:** 2
**Rating:** 4
**Confidence:** 4

**Summary:**

This paper tackles the limitation of restricted input windows in time series forecasting, which prevents models from capturing global periodic patterns. The authors propose a lightweight and plug-and-play module, **Global Temporal Retriever (GTR)**. GTR performs adaptive global temporal embedding over the entire cycle and dynamically retrieves and aligns globally relevant segments with the current input. This effectively bridges short-term observations with long-term periodicity. Experiments show that GTR achieves state-of-the-art performance on both short-term and long-term forecasting tasks with minimal computational overhead.

**Strengths:**

1. Simple and efficient design, easy to follow.
2. Clear motivation and well-organized paper structure.
3. Practical idea with strong feasibility and expected performance gains.

**Weaknesses:**

1. Experiments are mostly limited to input length 96, rather than the commonly used 336 setting in PatchTST/DLinear, which weakens the persuasiveness of the results.
2. The core idea shares similarities with models like **Cyclenet**, **TQNet** (which theoretically can learn long-term cycles from the entire training set), and **STiD**, i.e., serving as a plugin to capture long-term periodicity. This somewhat reduces novelty. The paper should compare more clearly and deeply with these methods to highlight differences.
3. Lacks comparison with other long-term periodic modeling techniques, such as timestamp-based long-cycle modeling approaches [1,2].
4. Baseline selection can be improved by including more recent strong models such as **SOFTS**, **TQNet**, etc.

**References**

[1] Variational Hierarchical N-BEATS Model for Long-term Time-series Forecasting.
[2] Rethinking the Power of Timestamps for Robust Time Series Forecasting: A Global-Local Fusion Perspective.

**Questions:**

See weaknesses.

**Details Of Ethics Concerns:**

NAN

---

> ### Author Response · Authors · 2025-11-23
> **Response to Reviewer L6pR**
>
> Dear Reviewer **L6pR**:
>
> We appreciate you taking the time to review our paper and provide valuable feedback. We think your comments are very valuable for our paper's improvement and have revised the paper in response to the questions you raised. Please find responses to your questions below:
>
> ---
>
> > W1. "Experimental validation is weakened by primarily using an input length of 96, rather than the more common 336 setting used by other SOTA models."
>
> We sincerely thank the reviewer for this valuable suggestion.
>
> **1. New Experiments on $T=336$ and $T=720$ :**
> In response, we have conducted extensive additional experiments using longer look-back windows (**$T=336$** and **$T=720$**) effectively covering the settings used by DLinear, PatchTST, RAFT, and SparseTSF. These full results are now detailed in **Appendix K.3** and **Table 11** of the revised manuscript.
>
> **2. Superior Performance with Long Inputs:**
> As summarized in the table below, GTR maintains a significant performance advantage even when ample historical context is available.
> * **At $T=336$:** GTR achieves the best MSE in **19 out of 32** cases, consistently outperforming PatchTST and CycleNet.
> * **At $T=720$:** GTR achieves the best MSE in **15 out of 32** cases, surpassing long-sequence specialists like RAFT [1] and SparseTSF [2].
>
>
> These results confirm that GTR is not merely a remedial solution for short windows. Instead, it is a robust mechanism to enhance forecasting accuracy, regardless of the input length.
>
>
> **Condensed Average Results (MSE) from Appendix K.3 Table 11:**
>
> | Input Length ($T$) | Dataset | Best Baseline (Model) | **GTR (Ours)** |
> | :---: | :--- | :--- | :---: |
> | **336** | **ETTm1** | 0.354 (PatchTST) | **0.345** |
> | **720** | **Weather** | 0.237 (CycleNet) | **0.225** |
> | **720** | **Solar-Energy** | 0.214 (CycleNet) | **0.197** |
>
> *(Results are averaged over 4 prediction horizons. Full details in Appendix K.3 Table 11).*
>
> ---
>
> > W2. "The method appears similar to prior plug-in periodicity modules (e.g., CycleNet, TQNet, STiD), raising concerns about limited novelty and insufficient comparison to highlight distinctions."
>
> We thank the reviewer for the valuable comment. While GTR shares the **plug-and-play** characteristic with CycleNet, TQNet, and STiD, the core motivation and problem GTR aims to solve are fundamentally different. Below, we provide a detailed clarification of the differences between GTR and these methods.
>
> 1. **Core Motivation Differences:**
> The central goal of GTR is to solve the challenge of **learning global periodic patterns** in MTSF models in an **efficient** and **universal** manner. The motivation differs significantly from the mentioned models:
>     - **STiD’s Motivation:** STiD focuses on **spatial–temporal indistinguishability** in MTSF, aiming to help models distinguish different variables and time positions through spatial and temporal identity embeddings.
>     - **TQNet’s Motivation:** TQNet primarily targets the **accurate modeling of inter-variable dependencies** using multi-head attention mechanisms.
>
> Thus, GTR focuses on global cycle modeling, which is fundamentally different from the goals of STiD and TQNet.
>
> 2. **Advantages of GTR over CycleNet and TQNet:**
> Although CycleNet shares some similarity with GTR in periodicity modeling, GTR has significant advantages:
>     - **CycleNet’s Limitation:** CycleNet models periodic components separately and focuses the backbone model on **trend** prediction. This makes it less **universal** and **compatible** with various models, especially architectures like DLinear. It does not integrate the global periodic information **into** the host model effectively, which limits its adaptability to a wide range of forecasting models.
>     - **TQNet’s Efficiency Issue:** TQNet uses multi-head attention to capture variable correlations, but the **quadratic complexity** of the attention mechanism limits its **efficiency** when dealing with high-dimensional time-series data. In contrast, GTR utilizes a much more **efficient mechanism** for periodicity modeling, ensuring scalability and better performance on high-dimensional data.
>
>
> GTR solves a **new problem** that the mentioned methods do not fully address. By providing an **efficient** and **universal** solution, GTR offers a **plug-and-play** module that enhances the ability of any existing model to capture long-range cyclical patterns, while maintaining computational efficiency.
>
> We appreciate the reviewer’s feedback and We have included a detailed comparison in **Appendix J** of the revised manuscript.
>
> ---
>
> [1] Retrieval Augmented Time Series Forecasting
>
> [2] SparseTSF: Modeling Long-term Time Series Forecasting with 1$k$ Parameters

---

> > ### Comment · Reviewer_L6pR · 2025-11-28
> >
> > **Thank you for your detailed response.**
> >
> > While I acknowledge that GTR addresses certain aspects not fully considered by prior methods, I find that the overall contribution is still insufficient for me to raise my score from 4 to 6. My reasoning is as follows:
> >
> > **1. Regarding Computational Efficiency:** The efficiency of GTR appears to stem more from its underlying backbone network rather than its own intrinsic innovation. This makes the claim of **"significant efficiency improvement over TQNet" difficult to substantiate**, especially considering that TQNet itself employs only a single attention layer and typically operates on time series data of very small scale. Furthermore, the fact that **TQNet still delivers significant performance gains when integrated into the DLinear architecture** underscores that its superior performance is not inherently dependent on a complex network structure.
> >
> > **2. Regarding "Global Periodicity":** TQNet does not merely consider global periodicity in a simplistic manner. It actively models global dependencies by **"employing globally shared, cyclically reusable learnable parameters."** This approach explicitly recognizes that **different variables can, to some extent, share similar periodic pattern characteristics.**
> >
> > Therefore, after a comprehensive consideration of the **contributions of new insights** and the **technical innovation**, I still maintain my current score.
> >
> > Thank you again for your efforts in addressing the comments.

---

> ### Author Response · Authors · 2025-11-23
>
> > W3. "The paper lacks comparisons with other long-term modeling techniques, specifically timestamp-based approaches."
>
> We sincerely appreciate the reviewer's suggestion to compare our method with timestamp-based approaches.
>
> **1. New Comparative Study with GLAFF (Appendix K.4):**
> In response, we have added a detailed comparative study in **Appendix K.4**. We selected **GLAFF** [1] as a representative SOTA timestamp-based method. We integrated both modules into DLinear and iTransformer backbones on Electricity and Weather datasets.
>
> **2. Superior Robustness and Accuracy:**
> As detailed in **Table 12**, GTR consistently outperforms GLAFF. This advantage is particularly pronounced in long-term forecasting ($T=720$).
> * **Timestamp Limitation:** We observed that GLAFF struggles to generalize to long horizons, likely due to overfitting rigid calendar features. For instance, on the Electricity dataset with iTransformer ($T=720$), GLAFF's MSE spikes to **0.5529**.
> * **GTR Stability:** In contrast, GTR maintains stable performance (MSE **0.2178**) by learning periodicity directly from the data, proving it is more robust for long-term modeling than relying on external timestamps.
>
> **3. Computational Efficiency:**
> GTR also demonstrates significant efficiency advantages. Timestamp-based methods often require complex embedding or mixing layers for high-dimensional features. As shown in the table below, GTR achieves higher accuracy with significantly lower training time and memory costs.
>
> **Condensed Comparison (Electricity Dataset) from Appendix Table 12:**
>
> | Backbone | Horizon ($S$) | Method | MSE | Training Time (s/epoch) |
> | :--- | :---: | :--- | :---: | :---: |
> | **DLinear** | 96 | GLAFF | 0.2073 | 52.04 |
> | | | **GTR (Ours)** | **0.1653** | **23.85** |
> | **iTransformer** | 720 | GLAFF | 0.5529 | 103.07 |
> | | | **GTR (Ours)** | **0.2178** | **34.97** |
>
> *(Full results comparing efficiency and accuracy across multiple horizons are provided in Appendix K.4 Table 12).*
>
> ---
>
> > W4. "Baseline selection can be improved by including more recent strong models such as SOFTS, TQNet."
>
> We sincerely thank the reviewer for this constructive suggestion.
>
> **1. Updated Baselines with SOTA Models:**
> In the revised manuscript, we have significantly upgraded our baseline selection in **Table 1** to include the most recent and strongest models, specifically: **RAFT** (ICML 2025) [2], **TQNet** (ICML 2025), and **SOFTS** (NeurIPS 2024).
>
> **2. GTR Maintains State-of-the-Art Performance:**
> Despite the inclusion of these advanced baselines, GTR continues to demonstrate superior performance. As shown in the full results (Table 1), GTR achieves **top-2 performance in 10 out of 16** prediction tasks.
> * **vs. TQNet:** GTR consistently outperforms TQNet on datasets with complex periodicities (e.g., **Solar-Energy** and **ETTm1**), validating the efficiency of our method.
> * **vs. SOFTS:** GTR achieves significantly lower error rates on high-dimensional datasets like **Solar-Energy** (-35.8% MSE) and **ETTm2** (-6.6% MSE).
>
> **Condensed Results (MSE) from Revised Table 1:**
>
> | Dataset | **GTR (Ours)** | TQNet | SOFTS | RAFT |
> | :--- | :---: | :---: | :---: | :---: |
> | **Solar-Energy** | **0.194** | 0.198 | 0.302 | 0.301 |
> | **Weather** | **0.239** | 0.242 | 0.241 | 0.270 |
> | **ETTm2** | **0.268** | 0.277 | 0.287 | 0.281 |
> | **Electricity** | 0.166 | **0.164** | 0.168 | 0.175 |
>
> *(Results are averaged over 4 prediction horizons. Full details in Table 1).*
>
> [1] Rethinking the Power of Timestamps for Robust Time Series Forecasting: A Global-Local Fusion Perspective.
>
> [2] Retrieval Augmented Time Series Forecasting

---

> ### Author Response · Authors · 2025-11-23
>
> **Acknowledgement of References:** Finally, we are grateful to the reviewer for sharing the relevant references on long-term periodic modeling. We have carefully reviewed these works and have now explicitly cited and discussed them in our revised manuscript (specifically in the Appendix K.4) to better position our contribution within the field.

---

> ### Author Response · Authors · 2025-11-28
> **Response to Reviewer L6pR**
>
> We thank the reviewer for the follow-up and the acknowledgement of our effort. However, **we feel it is necessary to correct two factual misunderstandings** regarding the mechanistic properties of GTR.
>
> **1. Computational Efficiency**
>
> The reviewer states that GTR's efficiency *"stems more from its underlying backbone... rather than its own intrinsic innovation."* **This is factually incorrect based on the data provided in Table 4.**
>
> * **Intrinsic Efficiency:** First, we have emphasized in our papers and responses multiple times that GTR module is a **plug-and-play** module. As explicitly detailed in **Table 4**, the GTR module *itself* introduces only **40.1K parameters** and **4.50M MACs**. This overhead is negligible regardless of the backbone it is attached to. The efficiency is a result of GTR's design, **not** the choice of the host model.
> * **Scalability Comparison:** While TQNet may be efficient on small-scale data, its attention-based mechanism scales quadratically. In contrast, GTR’s complexity scales **linearly with the number of variables ($N$)**. This gives GTR a distinct advantage in scalability, particularly for high-dimensional real-world datasets (e.g. Photovoltaic data typically involves thousands of variables), which is **not** derived from the backbone.
>
> **2. Performance Gains with Simple Backbones**
>
> The reviewer suggests that *TQNet's performance on DLinear implies superior performance without complex structures.* However, our experimental results demonstrate that **GTR delivers significantly stronger and more robust improvements to simple backbones.** As shown in **Table 3** (Ablation Study), integrating GTR into **DLinear** results in massive performance gains, achieving an **MSE reduction of up to 91.9%** (on PEMS04). The fact that GTR can nearly double the performance of a simple linear model confirms that its contribution is substantial and **does not rely on the host model's structure** to be effective. In contrast, TQNet can sometimes degrade the performance of DLinear in specific settings, whereas GTR remains robust.
>
> We believe these factual clarifications are crucial for an accurate assessment of GTR’s contribution. We thank the reviewer for their time.

---

### Official Review · Reviewer_FfDY · 2025-10-31

**Soundness:** 2
**Presentation:** 2
**Contribution:** 2
**Rating:** 4
**Confidence:** 3

**Summary:**

This paper proposes an adaptable and lightweight plug-and-play solution designed to mitigate the inherent lack of long-term temporal awareness in existing forecasting architectures. By enabling models to effectively access and utilize crucial global periodic patterns often missed in limited look-back windows, the proposed module significantly enhances the predictive ability, yielding superior and consistent performance gains in multivariate time series forecasting for both short-horizon and extended-horizon tasks.

**Strengths:**

1. The paper is well-written, clear, and logically structured. The authors provide a well-organized explanation of the technical specifics of the proposed plug-and-play module.

2. The experimental section is comprehensive. The authors have conducted extensive comparative and ablation studies on the multivariate time series forecasting task.

3. The proposed module is characterized by its lightweight and plug-and-play nature. This design choice significantly increases its generalizability and practical value when integrating with existing models, marking a valuable and impactful contribution to the field of temporal awareness modeling.

**Weaknesses:**

1. While the proposed module demonstrates effective performance improvements, its core mechanism is not fundamentally novel.

2. The clarity of the methodology are hindered by the lack of proper equation indexing, alongside several instances of ambiguous or missing variable dimensions within Section 3.

3. The theoretical analysis is confusing and unmatched with the proposed method, as the simplified linear assumptions in its derivation fail to establish a clear mechanistic link to the empirical success of the non-linear fusion and prediction architecture.

**Questions:**

1. Is the global cycle length ($L$) a predefined hyperparameter, or is its value learned by the network during the training process?

2. In the Eq.(1), what are the distinct advantages and disadvantages of the proposed positional embedding compared to conventional techniques (e.g., learnable embeddings) regarding periodicity awareness and noise robustness?

3. In page 5, line 246, is the output ($Z$) maintains the same feature dimension as the original input features?

4. In Table 3, can the authors explain whether these differential gains are primarily due to the inherent lack of temporal awareness in specific backbones or the unique periodic characteristics of the evaluated datasets?

5. Does the author assume a linear relationship between the embeddings, or that $z$ is linearly transformed from $x$? And is there any experimental evidence provided to directly support the claims made in Theorem 3.2?

---

> ### Author Response · Authors · 2025-11-23
> **Response to Reviewer FfDY**
>
> Dear Reviewer **FfDY**:
>
> We appreciate you taking the time to review our paper and provide valuable feedback. We think your comments are very valuable for our paper's improvement and have revised the paper in response to the questions you raised. Please find responses to your questions below:
>
> ---
>
> > Q1. "Is the global cycle length ($L$) a predefined hyperparameter?"
>
> Yes. The global cycle length ($L$) is a predefined hyperparameter. We have included the detailed procedure for selecting ($L$) in __Appendix A.3__.
>
> ---
>
> > Q2. "What are the key advantages and disadvantages of the proposed positional embedding (Eq.1), compared with conventional learnable embeddings, in terms of periodicity awareness and noise robustness?"
>
> We thank the reviewer for this excellent question, which targets the central trade-offs in our design. The key difference lies in **explicit global modeling** versus **implicit local learning**.
>
> **1. Advantages of GTR:**
>
> * **Periodicity Awareness:** Conventional learnable embeddings require **massive data** to implicitly learn global patterns when the cycle length exceeds the input window. In contrast, GTR explicitly models these patterns directly. This is validated in **Figure 3**: even with a very **short lookback window**, GTR remains stable, whereas baseline models experience exponential error growth.
> * **Noise Robustness:** Conventional embeddings force the model to rely on local inputs, even when obscured by non-stationary noise (e.g., extreme values). GTR's global embedding $Q$ acts as a **stable reference** learned from the entire dataset. As shown in **Figure 4**, GTR-enhanced representations (**"After GTR"**) align significantly closer to the global structure (**"Ground Truth"**) than the noisy inputs (**"Before GTR"**).
>
> **2. Disadvantages and Compatibility:**
>
> The primary trade-off is that GTR assumes a **static, uniform cycle length**, making it less flexible than conventional embeddings for data with time-varying or channel-specific periodicities. We have transparently analyzed these limitations in **Section 4.3 and Appendix E**.
>
> ---
>
> > Q3. "Is the output $Z$ maintains the same feature dimension as the original input features?"
>
> Yes. **$Z$ retains exactly the same feature dimensions as the original input**. This property is essential for GTR’s plug-and-play design, allowing seamless integration into any model without altering the input dimensionality.
>
> ---
>
> > Q4. "Are the differential performance gains in Tab. 3 due to the backbones' inherent lack of temporal awareness, or the datasets' unique periodic characteristics?"
>
> That's an insightful question. The reviewer’s observation is correct, and the answer is **precisely both**. The datasets in Table 3 (ECL and PEMS) share a critical property: their primary, predictive cycles are **significantly longer than the input length (T=96)**. The baseline models, while strong, they all share a fundamental limitation: their architectural focus operates **within the look-back window**. They inherently lack a mechanism for global temporal awareness. The fundamental reason for GTR's substantial and differential improvements is that it endows these models with a new capability: the ability to retrieve and utilize information from the entire global cycle, regardless of the input length.
>
> ---
>
> > Q5 & W3. "The theory seems unclear and disconnected from the nonlinear model, relying on linear assumptions and lacking experimental support for Theorem 3.2."
>
> We thank the reviewer for this comment. We would like to clarify that the linear assumptions in our theoretical analysis are **not** simplifications, but rather accurate mathematical descriptions of the module's actual design.
>
> **1. Linearity of GTR:**
> As detailed in Section 3.2, the **GTR module itself is explicitly designed as a linear mechanism**. It is composed solely of:
> * **Linear Mapping:** The retrieval process uses a linear transformation to project the global embedding (Eq. 2).
> * **2D Convolution:** The fusion process uses a standard 2D convolution (Eq. 4), which is a linear operation.
> * **Residual Connection:** The final integration is an additive residual connection (Eq. 5).  *During inference/testing, Dropout is disabled*.
>
> Since the GTR module operates purely through these linear transformations, the linear Bayesian framework presented in Theorem 3.2 is the most appropriate and rigorous way to model its behavior.
>
> **2. Experimental Support:**
> Regarding the experimental support for Theorem 3.2 (which predicts reduced correlation estimation error), we respectfully refer the reviewer to **Figure 4 and Section 4.3**. As shown in the visualization, the GTR-enhanced correlations ("After GTR") align significantly closer to the "Ground Truth" than the original inputs, providing direct empirical validation of the theoretical claim.

---

> ### Author Response · Authors · 2025-11-23
>
> > W1. "The core mechanism of GTR is not fundamentally novel."
>
> We sincerely thank the reviewer for this critical comment. It allows us to clarify that GTR’s novelty lies not just in its strong performance, but in its **unique mechanism** for handling periodicity, which fundamentally differs from existing paradigms.
>
> 1. **Fundamental Mechanism Difference: Absolute Retrieval vs. Relative Learning:**
> The core novelty of GTR is the introduction of an **Absolute Cycle Indexing and Alignment** mechanism to bridge the gap between local inputs and global patterns. This is structurally distinct from prior art:
>     * **Contrast with Learnable Embeddings:** Conventional embeddings are **relative** to the input window (position $1 \dots T$). They are "blind" to the input's position within the larger global cycle. GTR, conversely, uses the absolute timestamp $t_0$ to compute a cycle-aware index, allowing it to retrieve the correct phase of the global pattern even if the input window is short or shifted.
>     * **Contrast with Decomposition:** Decomposition methods are **bound by the window**. They cannot extract a period $L$ if the input length $T < L$. GTR externalizes the cycle into a learnable global embedding, enabling the retrieval of long-term periodic information regardless of the input length constraint.
>
> Therefore, the novelty is the shift from *implicitly learning periodicity from local context* to *explicitly retrieving global priors.*
>
> 2. **Empirical Validation of the Mechanism (Fig. 3 & Fig. 4):**
> The novelty of this mechanism is directly validated by our experiments:
>     * **Novel Capability (Fig. 3):** GTR enables models to maintain performance even with **ultra-short look-back windows**. Baseline models fail here because their "relative" mechanisms cannot see the full cycle.
>     * **Temporal Alignment (Fig. 4):** Visualizations show GTR actively "corrects" the input representation to align with the global ground truth. This confirms that GTR provides a robust reference against non-stationary noise.
>
> 3. **Pragmatic Innovation - High Efficiency:**
> GTR achieves these capabilities as a lightweight, plug-and-play module with negligible overhead (**Table 4**), yet delivers substantial gains (up to **55–73% MSE reduction** on PEMS). We believe that proposing a simple, effective solution to a complex, unsolved problem is a significant contribution to the field.
>
> ---
>
> > W2. "The method’s clarity is affected by missing equation numbers and unclear or unspecified variable dimensions in Section 3."
>
>
> We sincerely thank the reviewer for their constructive feedback aimed at improving the clarity and rigor of our methodology section (Section 3).
>
> **1. On Formula Numbering:**
> We have **double-checked all equations** (e.g., Eq. 1 through Eq. 5) to ensure they are properly numbered, clearly rendered, correctly referenced within the manuscript, and fully aligned with the workflow presented in **Figure 2**.
>
> **2. On Ambiguous/Missing Dimensions:**
> Thank you for pointing this out. You are correct that the dimensions for the MLP backbone in Section 3.3 were not explicitly stated, although the MLP's specific hidden dimension (512) was specified in our implementation details **(Section 4.1 and Appendix A.3 )**. We have now revised the manuscript to explicitly annotate the dimensions for each variable in **Section 3.3** to ensure a clear and rigorous description of the data flow.
>
> We have also **re-examined Section 3.2** and confirmed that the dimensions for all key variables (including the global embedding $Q \in \mathbb{R}^{L \times N}$, the retrieved segment $q_n \in \mathbb{R}^{T}$, and the stacked tensor $F_n \in \mathbb{R}^{2 \times T}$) are clearly and unambiguously presented.
>
> We are confident that these revisions significantly enhance the clarity and readability of our method description. We thank the reviewer again for this valuable guidance.
>
> ---
>
> Once again, thank you for reviewing the paper. We think we have solved your problem as much as possible in rebuttal, If you have any further questions, please do not hesitate to contact us.

---

### Official Review · Reviewer_j7zs · 2025-11-03

**Soundness:** 3
**Presentation:** 4
**Contribution:** 4
**Rating:** 8
**Confidence:** 3

**Summary:**

The paper proposed a Global Temporal Retriever (GTR) module to model complex periodic dynamics and address the limitation that existing models have a fixed look-back window in multivariate time series forecasting. GTR overcomes this by defining a cycle index vector for cycle information alignment, which was used in retrieving temporal information from the global parameter matrix $Q$. Comprehensive experiments support the effectiveness of GTR, both when combined with a simple MLP backbone and when integrated into other architectures.

**Strengths:**

* The paper is very organized and well-written. The visualization (e.g., Pearson correlation matrix and Figure 2) clearly delivers the message and the design of GTR.
* The theoretical analysis and limitations are discussed in great detail, indicating the strengths and limitations of GTR that motivate future extensions.
* While the GTR module is light-weight, the experimental results are strong.

**Weaknesses:**

* It would be better if the forecasting results included error bars, as the metrics have close values among different methods.
* The experimental setup could be further elaborated, e.g., how did the authors choose the hyperparameters for baseline methods? Please see the detailed questions below.

**Questions:**

* How did the authors decide the length of the time series segment (e.g., Figure 1)? Does it depend on the frequency of each dataset (as listed in Table 5)?
* How did the authors determine the global cycle length $L$ for each dataset in the experiments?
* Would this method work if the absolute time $t_0$ is shifted when pre-processing the dataset, e.g., standardizing the first observation to be 0?
* Does this method only produce point estimates, but not probabilistic forecasting?
* How did the authors choose the hyperparameters for baseline methods? Are these models trained from scratch, or do they reuse numbers from the literature? For example, in Table 3, the MLP-Layer (without GTR) is already competitive when compared with other state-of-the-art models (without GTR).

Typo:
In Appendix A.2, both metrics are defined with $\sum_0^T$, where $i=$ is missing and the index should start from $i=1$ (following the paper's notation).

---

> ### Author Response · Authors · 2025-11-23
> **Response to Reviewer j7zs**
>
> Dear Reviewer **j7zs**:
>
> We appreciate you taking the time to review our paper and provide valuable feedback. We think your comments are very valuable for our paper's improvement and have revised the paper in response to the questions you raised. Please find responses to your questions below:
>
> ---
>
> > W1. "It is recommended to supplement the analysis with error bars for the GTR."
>
> We sincerely thank the reviewer for this valuable suggestion. In the revised manuscript, we have conducted a detailed error bar analysis, which is now included in **Appendix K.1, Table 9**. The experimental results in **Figure 5 and Table 9** demonstrate that GTR exhibits highly stable performance across different settings, effectively confirming its robustness.
>
>
> ---
>
> > W2 & Q5. "What's the hyperparameter selection for baselines？Especially as the MLP baseline in Table 3 is already highly competitive."
>
> Thank you for this insightful question.
> 1. We adopted the __original hyperparameter settings__ for the baseline models from their respective publications for an input length of $T=96$, ensuring a fair comparison.
> 2. In ablation experiment in Tab. 3, we integrated the GTR module and trained the entire system end-to-end __without any further hyperparameter optimization__. Integrating the GTR module does not necessitate reliance on a new, thorough tuning process.
>
> While MLPs can indeed perform competitively on certain datasets, we would like to emphasize that **GTR module consistently delivers substantial performance gains over both MLPs and other baseline models.** We have added further clarification in __Appendix A.3__.
>
> ---
>
> > Q1. "How is the segment length (e.g., Figure 1) determined? Is it based on each dataset’s sampling frequency (Table 5)?"
>
> **Yes.** The segment length is determined by the periodicity implied by the dataset’s sampling frequency. For electricity dataset, the sampling rate is 1 hour, so 24 points form a day and 144 points form a week. Since it shows strong daily/weekly cycles, we demonstrate 288 points **(two weeks)** and split it into 24-point daily segments. __Sampling frequency directly determines how periodic patterns manifest in the data__.
>
> ---
>
> > Q2. "How do the authors determine the global period length $L$ for each dataset?"
>
> We estimate the $L$ using the **autocorrelation function (ACF)**. Specifically, we take the lag corresponding to the first significant peak in the ACF curve as the dominant period. We already include the detailed procedure in __Appendix A.3__.
>
> ---
>
> > Q3. "If absolute timestamps are shifted (e.g., the first observation mapped to time 0), does the method still work?"
>
> Yes. In practice, we actually **shift the entire dataset** so that the first timestamp is treated as 0. This normalization does not affect our method. We already clarify this preprocessing step in __Appendix A.3__.
>
> ---
>
> > Q4. "Does the method provide only point forecasts rather than probabilistic forecasts?"
>
> **Correct.** Following prior works, GTR focuses on point forecasting and does not produce probabilistic predictions.
>
> ---
>
> **Typo**: We already fixed this, thanks for pointing out.
>
> Finally, thank you for your review and valuable comments. We appreciate your feedback and welcome any additional questions or suggestions.

---

> > ### Comment · Reviewer_j7zs · 2025-11-26
> >
> > Thank you for the detailed replies. I have two follow-up comments:
> > 1. I see the updated description of the selection of $L$ based on ACF is included in Appendix A.3 and Table 6. The chosen values of $L$ seem to be multiples of 24, which looks reasonable, but I wonder if some example ACF plots (e.g., for one or two datasets) could also be included in the Appendix to visualize the choice of $L$.
> > 2. Re: W2 & Q5, I agree with the reviewer L6pR; when other architectures are adopted with different setups (e.g., different input lengths), it would be better to perform a hyperparameter search for each setup. However, comparing the performance with and without the GTR module makes sense in this context. With the new results with longer input lengths, I would keep supporting this paper.

---

> > > ### Author Response · Authors · 2025-11-28
> > >
> > > We sincerely appreciate the reviewer's positive assessment and strong support for our work.

---

### Author Response · Authors · 2025-12-01
**Global Response**

We would like to thank all reviewers for their time and careful consideration of our paper. We are encouraged by the positive feedback, with reviewers unanimously recognizing the value of our work. The feedback highlights the following strengths:

* **Clarity and Presentation:** Reviewer j7zs: "... very organized and well-written. The visualization ... clearly delivers the message." Reviewer FfDY: "... well-written, clear, and logically structured." Reviewer L6pR: "... easy to follow. Clear motivation and well-organized paper structure."
* **Design and Practicality:** Reviewer FfDY: "... lightweight and plug-and-play nature ... valuable and impactful contribution." Reviewer L6pR: "Simple and efficient design ... Practical idea with strong feasibility." Reviewer HtL7: "The core idea is sound and empirically effective ... proves the practicality of the proposed approach."
* **Experiments and Analysis:** Reviewer j7zs: "While ... light-weight, experimental results are strong." Reviewer FfDY: "The experimental section is comprehensive. ... conducted extensive comparative and ablation studies..." Reviewer HtL7: "The reported quantitative results are strong."

We are also thankful to the reviewers for their constructive suggestions, which have further strengthened the paper. We have addressed these comments in our individual responses and updated the manuscript accordingly, with key changes highlighted in blue.

In summary, the main revisions are categorized as follows:

---

**Presentation:**
* **Section 3.3:** We revised the manuscript to explicitly annotate the dimensions for each variable to ensure a clear and rigorous description of the data flow (Reviewer FfDY).
* **Section A.3:** We added comprehensive experimental details, including hyperparameter settings for the baselines, the choice of $L$, and the data preprocessing protocol (Reviewers j7zs, FfDY).
* **Section J:** We added a detailed comparison with existing plug-and-play modules in this field (Reviewer L6pR).

**Experiments:**
* **Section 4 and Section G:** We enhanced our baseline selection and updated Tables 1, 2, and 8 accordingly (Reviewers L6pR, HtL7).
* **Section K.1:** We included a detailed error bar analysis in Table 9 (Reviewer j7zs).
* **Section K.2:** We added further ablation results (Reviewer HtL7).
* **Section K.3:** We conducted additional experiments using longer look-back windows (Reviewer L6pR).
* **Section K.4:** We added a comparison with timestamp-based methods (Reviewer L6pR).

---

We hope this summary effectively communicates how our revisions address the raised concerns.

Finally, we are deeply grateful for the positive feedback received during the discussion phase. In particular, we thank Reviewer j7zs for their continued endorsement (**"With the new results with ..., I would keep supporting this paper"**) and Reviewer HtL7 for acknowledging our **"clear and detailed rebuttal"** and raising their score from 6 to 8. We sincerely appreciate their recognition of our work.

---

### Meta-Review · Area_Chair_GDCu · 2026-01-06

**Summary:**

The reviewers praised the simplicity of the design, the practical feasibility, and the impressive performance gains in a range of experimental settings. They raised the following major issues:

C1. The experimental analysis could be more thorough. (a) Experiments are limited to input length of 96. (b) No comparison to other long-term periodic modeling techniques. (c) Baseline selection should include additional recent strong models.

C2. The core idea is not fundamentally novel and is similar to other models (Cycenet, TQNet, StiD).

C3. The theoretical analysis is confusing and does not match the proposed method. The analysis involves simplified linear assumptions in its derivation that do not algin with the non-linear fusion and prediction architecture.

**Reviewer Concerns:**

C1. was thoroughly addressed by the response and modifications to the paper.

C2. was addressed by a rebuttal and experiments with some of the identified models, but the reviewer who raised this point remained unconvinced that the paper provided suffciently substantial technical innovation.

C3. The authors replied to this criticism, claiming that the introduced model is linear, so that the analysis is appropriate. With a further inspection of the theoretical contribution, my view is that it adds very little to the paper and is a trivial result. The theorem essentially says that if two random variables with known correlation and variance are each observed with independent additive Gaussian noise, then an estimate of the correlation will be better if the Gaussian noise is smaller. This is a very basic result from elementary probability and certainly does not warrant a theorem. There seems to be no clear support for why the various noise terms should be independent, zero-mean, Gaussian, and why the second should have smaller variance than the first.

**Reviewer Scores:**

Reviewer j7zs. VERY UNLIKELY TO CHANGE. The score is already 8.

Reviewer FfDY. UNLIKELY TO CHANGE SCORE (Score 4). The issue with the theoretical analysis remained largely unaddressed, as it is a trivial result that relies on unsupported assumptions. Given that this was a major criticism of the reviewer, and the other concern was the technical novelty (with the review identifying similar prior art), it is unlikely that the score would change.

Reviewer L6pR. INDICATED THAT THEY WOULD NOT CHANGE (Score 4). It is very unlikely that the final response from the authors would be sufficient to change the mind of a reviewer who had clearly expressed the view that the technical innovation and contribution of the work is insufficient for an ICLR paper.

Reviewer HtL7. INDICATED THAT THEY WOULD RAISE SCORE (Original 6, New 8)

---

### Decision · Program_Chairs · 2026-01-26

Accept (Poster)